# Sign Operator for Coping with Heavy-Tailed Noise: High Probability Convergence Bounds with Extensions to Distributed Optimization and Comparison Oracle

## Abstract

The growing popularity of AI optimization problems involving severely corrupted data has increased the demand for methods capable of handling heavy-tailed noise, i.e., noise with bounded $\kappa$-th moment, $\kappa \in (1,2]$. For the widely used clipping technique, effectiveness heavily depends on the careful tuning of clipping levels throughout training. In this paper, we demonstrate that using only the sign of the input, without introducing additional hyperparameters, is sufficient to cope with heavy-tailed noise effectively. For smooth non-convex functions, we prove that SignSGD achieves optimal sample complexity $\tilde{O}\left(\varepsilon^{-\frac{3\kappa-2}{\kappa-1}}\right)$ with high probability for attaining an average gradient norm accuracy of $\varepsilon$. Under the assumption of symmetric noise, we use SignSGD with Majority Voting to extend this bound to the distributed optimization or reduce the sample complexity to $\tilde{O}(\varepsilon^{-4})$ in the case of a single worker with arbitrary parameters. Furthermore, we explore the application of the sign operator in zeroth-order optimization with an oracle that can only compare function values at two different points. We propose a novel method, MajorityVote-CompsSGD, and provide the first-known high-probability bound $\tilde{O}(\varepsilon^{-6})$ for the number of comparisons under symmetric noise assumption. Our theoretical findings are supported by the superior performance of sign-based methods in training Large Language Models.

[1]Anonymous Institution, Anonymous City, Anonymous Region, Anonymous Country. Correspondence to: Anonymous Author <anon.email@domain.com>.

Preliminary work. Under review by the International Conference on Machine Learning (ICML). Do not distribute.

## 1. Introduction

**Problem statement.** Consider the stochastic optimization problem of a smooth non-convex function $f : \mathbb{R}^d \to \mathbb{R}$:

$$\min_{x \in \mathbb{R}^d} f(x) := \mathbb{E}_{\xi \sim \mathcal{S}}[f(x, \xi)], \tag{1}$$

where random variable $\xi$ can only be sampled from an unknown distribution $\mathcal{S}$. The gradient oracle gives unbiased gradient estimate $\nabla f(x, \xi) \in \mathbb{R}^d$. For example, in machine learning, $f(x, \xi)$ can be interpreted as a loss function on a sample $\xi$ (Shalev-Shwartz & Ben-David, 2014).

The most popular approach for solving (1) is Stochastic Gradient Descent (SGD) (Robbins & Monro, 1951):

$$x^{k+1} = x^k - \gamma_k \cdot g^k, \quad g^k := \nabla f(x^k, \xi^k).$$

For non-convex functions, the main goal of stochastic optimization is to find a point with small gradient norm.

Huge success of stochastic first-order methods in rapidly developing neural networks field (Bottou, 2012; Kingma & Ba, 2014) has sparked numerous works studying SGD under various assumptions on corrupting noise induced by the randomness $\xi$. The first bounds in expectation for the sample complexity were derived for sub-Gaussian noise (Nemirovski et al., 2009) and for noise with bounded variance (BV) (Ghadimi & Lan, 2013).

Due to expensive single run training of large deep learning models (Davis et al., 2021), more informative *high probability (HP)* bounds have gained even more attention than bounds in expectation describing methods behavior over several runs. HP bounds provide convergence guarantees which hold true with probability at least $1 - \delta, \delta \in (0, 1)$. The bound in expectation can be reduced to the HP bound using the Markov's inequality, however, it leads to a dominant $1/\delta$ factor. HP bounds for SGD under Gaussian noise were obtained in (Li & Orabona, 2020) and had logarithmic $\log 1/\delta$ dependency. However, already under the BV noise, vanilla SGD achieves only $1/\sqrt{\delta}$ rate (Sadiev et al., 2023). Moreover, it was shown that BV assumption can not describe loss functions in modern deep learning problems. In Large Language Models (LLMs), the stochasticity tends to

have a rather *heavy-tailed (HT)* distribution (Simsekli et al., 2019; Zhang et al., 2020b; Gurbuzbalaban et al., 2021). It means that the noise has bounded $\kappa$-th moment for some $\kappa \in (1, 2]$, i.e., $\mathbb{E}_\xi[\|\nabla f(x, \xi) - \nabla f(x)\|_2^\kappa] \leq \sigma^\kappa$. Desire to obtain better $\delta$-dependency and consider HT noise in HP bounds motivated development of more robust methods.

## 1.1. Related works

**Clipping.** The idea of clipping the norm of gradient estimate before SGD step demonstrates great empirical results (Pascanu et al., 2013; Goodfellow et al., 2016) and helps achieve $\log 1/\delta$ dependency under BV assumption (Nazin et al., 2019; Gorbunov et al., 2020).

The clipping operator is defined as $\text{clip}(g^k, \lambda_k) := \min\{1, \lambda_k/\|g^k\|_2\} \cdot g^k$ and can be applied not only non-convex minimization problems, but also to convex optimization, variational inequalities (Sadiev et al., 2023), non-smooth optimization (Zhang et al., 2020b), zeroth-order optimization (Kornilov et al., 2024), robust aggregation (Karimireddy et al., 2021), distributed optimization (Liu et al., 2022; Qin et al., 2025) and ensuring differential privacy (Andrew et al., 2021).

Let us list the latest results on the HP convergence of SGD with clipping, called ClipSGD, under HT noise assumption. First, for non-convex functions, the authors of (Zhang et al., 2020b) proved lower bounds for sample complexity in expectation. As shown in (Sadiev et al., 2023), ClipSGD with proper clipping levels and stepsizes achieves this lower bound with extra logarithmic factors on $\delta$ and accuracy. In a number of works, authors relax the HT assumption and consider only symmetric noises. This relaxation allows them to eliminate the dependency on $\kappa$ and break actual lower bounds. For example, in (Puchkin et al., 2024), the authors used the coordinate-wise median operator, which requires only a few noise samples to lighten its distribution so that it becomes BV. (Puchkin et al., 2024) proved convergence of ClipSGD combined with median operator for (strongly) convex functions as if $\kappa = 2$.

Despite clipping's effectiveness, it requires careful tuning of clipping levels whose optimal values depend on the iteration number and all characteristics of objective function and HT noise (Sadiev et al., 2023, Theorem 3.1). Hence, the community has proposed other robust modifications of SGD.

**Normalization.** A natural relaxation of clipping with profound level schedule is permanent normalization of gradient estimate, i.e, $\text{norm}(g^k) := \frac{g^k}{\|g^k\|_2}$. SGD with additional normalization is called NSGD (Hazan et al., 2015).

In early works devoted to NSGD, only BV noise and bounds in expectation were considered (Barakat et al., 2023; Yang et al., 2024). In (Liu et al., 2023; Cutkosky & Mehta, 2021), normalization was combined with clipping, which helped

cope with HT noise and obtain HP bounds.

Recently, HP convergence of vanilla NSGD was proved under HT noise in (Hübler et al., 2024). The authors showed that complexity of NSGD exactly matches the before mentioned lower bound from (Zhang et al., 2020b) without logarithmic accuracy factors and only with mild $\log 1/\delta$ dependency. Moreover, in experiments with sequence labeling via LSTM Language Models (Merity et al., 2017), normalization demonstrated better results than tuned clipping.

Unlike ClipSGD, NSGD requires large batchsizes for robust convergence. It can be fixed by replacing batching with momentum techniques, which keeps the same sample complexity (Cutkosky & Mehta, 2020). However, for methods with momentum, the convergence bounds are usually proved in expectation, and the HP bounds with logarithmic $\delta$ dependency has not been obtained yet. In experiments with VB noise, one can observe super-linear dependency on $\log \frac{1}{\delta}$ (Hübler et al., 2024).

**Sign operator.** There is one more promising modification of SGD which behavior under heavy-tailed noise has not yet been studied. Originally proposed in (Bernstein et al., 2018a) for distributed optimization, SignSGD takes only a sign of each coordinate of gradient estimate

$$x^{k+1} = x^k - \gamma_k \cdot \text{sign}(g^k).$$

There is one peculiarity in bounds for sign-based methods: they are proved w.r.t. the $\ell_1$-norm instead of smaller $\ell_2$-norm. As a consequence, additional $d$ dependent factors appear. Under BV noise, SignSGD achieves optimal sample complexity in expectation (up to $d$ factors). Similar to NSGD, SignSGD requires aggressive batching which can be substituted with momentum (Sun et al., 2023). The alternative solution is to add error feedback mechanism that additionally fixes the biased nature of sign operator (Seide et al., 2014; Karimireddy et al., 2019).

The main motivation of original SignSGD was communication effectiveness and empirical robustness in the distributed optimization (Bernstein et al., 2018b), since sending sign vector costs $O(d)$ operations. In theory, the effectiveness was proved only under additional assumptions on noise, e.g., symmetry and unimodality. Other applications and expansions of SignSGD are as follows: (Safaryan & Richtárik, 2021) proposed an updated theory for a wider class of noises in the distributed setup, (Liu et al., 2019a) generalized SignSGD to zeroth-order oracle, (Jin et al., 2020) studied federated learning and additional compression.

For all before-mentioned works, the results were obtained *in expectation and only for BV noise*. The HP bounds were obtained only in (Armacki et al., 2023; 2024), where the authors proposed a unified framework for theoretical analysis of online non-linear SGD. It includes a wide range of non-

linear gradient estimation transformations such as *clipping, normalization and sign operator.* However, in these works, the HT assumption was relaxed to symmetric noises only. The authors proved HP bounds which are arbitrarily close to the optimal ones for BV noise.

### 1.2. Contributions.

In our paper, we demonstrate that sign-based methods can handle heavy-tailed noise in zeroth- and first-order smooth non-convex optimization more effectively than clipping or normalization:

- We prove *the first high probability optimal bounds* $\tilde{O}\left((\sqrt{d}/\varepsilon)^{\frac{3\kappa-2}{\kappa-1}}\right)$ *under heavy-tailed noise* $\kappa \in (1,2]$ for SignSGD with mini-batching (Th. 1). For SignSGD with momentum, we generalize this bound in expectation (Th. 3).

- For symmetric heavy-tailed noise $\kappa \in (1,2]$, we combine SignSGD with majority voting and achieve an exact high-probability bound $\tilde{O}\left((\sqrt{d}/\varepsilon)^4\right)$ for the sample complexity (Th. 2).

- For the zeroth-order oracle which can only compare function values at two points, we propose a novel and simple MajorityVote-CompSGD method. We prove *the first high probability bound* $\tilde{O}((d/\varepsilon^2)^3)$ *for the comparison number under symmetric heavy-tailed noise* (Th. 4). For sum-type functions, we prove the bound $\tilde{O}\left((\sqrt{d}/\varepsilon)^{\frac{4\kappa-2}{\kappa-1}}\right)$ for the number of function calls under any heavy-tailed noise $\kappa \in (1,2]$.

- To validate our findings in real-world scenarios with heavy-tailed noise, we evaluate the sign-based methods on Transformer models, demonstrating their effectiveness in both pre-training LLaMA 130M on C4 dataset and zeroth-order fine-tuning RoBERTa on multiple NLP classification tasks.

**Notations.** The notation $\overline{1,n}$ is the set of natural numbers $\{1, 2, ..., n\}$. For a vector $x \in \mathbb{R}^d$, index $i \in \overline{1,d}$ returns its $i$-th coordinate $x = (x_1, \ldots, x_d)$. We define $\ell_p$-norm $p \in [1, +\infty)$ as $(\|x\|_p)^p := \sum_{i=1}^d |x_i|^p, x \in \mathbb{R}^d$. The notation $\langle x, y \rangle := \sum_{i=1}^d x_i y_i$ stands for the standard scalar product for $x, y \in \mathbb{R}^d$.

The sign operator $\text{sign}(\cdot)$ returns the sign of a scalar input and can be applied coordinate-wisely to a vector. The notation $\tilde{\mathcal{O}}$ hides logarithmic factors in failure probability $\delta$.

## 2. High probability bounds for sign-based methods under heavy-tailed noise

In this section, we present our novel convergence guarantees with high probability for existing sign-based methods for non-convex functions with heavy-tailed noise in gradient estimates. For each algorithm, we provide explicit optimal tuning for parameters. If function's smoothness constant and noise's characteristics are not given, we state the rates for arbitrary tuning. All proofs are located in Appendix C.

### 2.1. Assumptions

**Assumption 1** (Lower bound). *The objective function $f$ is lower bounded by $f^* > -\infty$, i.e., $f(x) \geq f^*, \forall x \in \mathbb{R}^d$.*

**Assumption 2** (Smoothness). *The objective function $f$ is differentiable and $L$-smooth, i.e., for the positive constant $L$*

$$\|\nabla f(x) - \nabla f(y)\|_2 \leq L\|x - y\|_2, \quad \forall x, y \in \mathbb{R}^d.$$

**Assumption 3** (Heavy-tailed noise in gradient estimates). *The unbiased estimate $\nabla f(x, \xi)$ has bounded $\kappa$-th moment $\kappa \in (1, 2]$ for each coordinate, i.e., $\forall x \in \mathbb{R}^d$:*

- $\mathbb{E}_\xi[\nabla f(x, \xi)] = \nabla f(x),$
- $\mathbb{E}_\xi[|\nabla f(x, \xi)_i - \nabla f(x)_i|^\kappa] \leq \sigma_i^\kappa, i \in \overline{1,d},$

*where $\vec{\sigma} = [\sigma_1, \ldots, \sigma_d]$ are non-negative constants. If $\kappa = 2$, then the noise is called bounded variance.*

### 2.2. SignSGD and its HP convergence properties

We begin our analysis with the simplest of sign-based methods, namely, SignSGD (Alg. 1) and prove a general lemma about its convergence with high probability.

---

**Algorithm 1** SignSGD

---

**Input:** Starting point $x^1 \in \mathbb{R}^d$, number of iterations $T$, stepsizes $\{\gamma_k\}_{k=1}^T$.
1: **for** $k = 1, \ldots, T$ **do**
2:     Sample $\xi^k$ and compute estimate $g^k = \nabla f(x^k, \xi^k)$;
3:     Set $x^{k+1} = x^k - \gamma_k \cdot \text{sign}(g^k)$;
4: **end for**
**Output:** uniformly random point from $\{x^1, \ldots, x^T\}$.

---

**Lemma 1** (SignSGD **Convergence Lemma**). *Consider lower-bounded $L$-smooth function $f$ (As. 1, 2) and HT gradient estimates (As. 4). Then Alg. 1 after $T$ iterations with constant stepsizes $\gamma_k \equiv \gamma$ achieves with probability at least $1 - \delta$ starting with $\Delta_1 = f(x^1) - f^*$:*

$$\frac{1}{T}\sum_{k=1}^T \|\nabla f(x^k)\|_1 \leq \frac{2\Delta_1}{T\gamma} + 16Ld\gamma \log(1/\delta) + 4\|\vec{\sigma}\|_1$$

$$+ 12\frac{d\|\nabla f(x^1)\|_1}{T} \log(1/\delta). \quad (2)$$

From the bound (2), one can derive the sample complexity for arbitrary parameters or calculate the optimal ones. In order to achieve accuracy $\varepsilon$, the noise $\|\vec{\sigma}\|_1$ have not to exceed $\varepsilon$. The first way to lower the noise is to use batching.

### 2.3. SignSGD **with batching**

---
**Algorithm 2** minibatch-SignSGD
---
**Input:** Starting point $x^1 \in \mathbb{R}^d$, number of iterations $T$, stepsizes $\{\gamma_k\}_{k=1}^T$, batchsizes $\{B_k\}_{k=1}^T$.

1: **for** $k = 1, \dots, T$ **do**
2:     Sample $\{\xi_i^k\}_{i=1}^{B_k}$ and compute gradient estimate
    $g^k = \sum_{i=1}^{B_k} \nabla f(x^k, \xi_i^k)/B_k$;
3:     Set $x^{k+1} = x^k - \gamma_k \cdot \text{sign}(g^k)$;
4: **end for**
**Output:** uniformly random point from $\{x^1, \dots, x^T\}$.

---

**Theorem 1** (**HP complexity for** minibatch-SignSGD). *Consider lower-bounded $L$-smooth function $f$ (As. 1, 2) and HT gradient estimates (As. 3). Then Alg. 2 requires the sample complexity $N$ to achieve $\frac{1}{T}\sum_{k=1}^T \|\nabla f(x^k)\|_1 \le \varepsilon$ with probability at least $1 - \delta$ for:*

*arbitrary tuning:* $T, \gamma_k \equiv \frac{\gamma_0}{\sqrt{T}}, B_k \equiv \max\{1, B_0 T\}$:

$$N = O\left( \frac{B_0(\Delta_1/\gamma_0 + Ld\gamma_0)^4}{\varepsilon^4} + \frac{1}{B_0}\left( \frac{\|\vec{\sigma}\|_1}{\varepsilon} \right)^{\frac{2\kappa}{\kappa-1}} \right), \tag{3}$$

*optimal tuning:* $T = O\left( \frac{\Delta_1 L_\delta d}{\varepsilon^2} \right), \gamma_k \equiv \sqrt{\frac{\Delta_1}{8L_\delta dT}}, B_k \equiv \max\left\{ 1, \left( \frac{16\|\vec{\sigma}\|_1}{\varepsilon} \right)^{\frac{\kappa}{\kappa-1}} \right\}$:

$$N = O\left( \frac{\Delta_1 L_\delta d}{\varepsilon^2} + \frac{\Delta_1 L_\delta d}{\varepsilon^2}\left( \frac{\|\vec{\sigma}\|_1}{\varepsilon} \right)^{\frac{\kappa}{\kappa-1}} \right), \tag{4}$$

*where $\Delta_1 = f(x^1) - f^*, L_\delta = L\log(1/\delta)$.*

### 2.4. SignSGD **with majority voting**

The second approach to noise reduction inherent to sign-based methods is majority voting.

**Majority voting.** As mentioned before, the original motivation of SignSGD was fast communication in the distributed optimization (Bernstein et al., 2018b; Jin et al., 2020). Consider one server and $M$ workers, each of which computes its own gradient estimate. The server receives signs of all estimates, aggregates them and sends back the updated estimate to the workers. In related works, various types of aggregation were studied, but the most effective one turned out to be majority voting. For sign vectors $\text{sign}(g_i^k), i \in \overline{1, M}$, each coordinate of the resulting update

vector is the majority of the received signs:

$$g^k = \text{sign}\left( \sum_{i=1}^M \text{sign}(g_i^k) \right). \tag{5}$$

To be effective, majority voting must decrease the noise of the aggregated update vector. This is achieved via showing that probability $\mathbb{P}\left[ \text{sign}(\nabla f(x^k)_j) \ne \text{sign}\left[ \sum_{i=1}^M \text{sign}(g_i^k)_j \right] \right]$ decreases with the growth of $M$. However, for arbitrary noise distributions, it does not hold true. Choosing the most frequent value from the sign sequence $\{\text{sign}(g_i^k)\}_{i=1}^M$ is actually $M$ Bernoulli trials. In these trials, the probability of choosing a correct answer grows only if the probability of failure of a single worker is less than $\frac{1}{2}$, i.e.:

$$\mathbb{P}\left[ \text{sign}(\nabla f(x^k)) \ne \text{sign}(g_i^k) \right] < \frac{1}{2}, \forall i \in \overline{1, M}. \tag{6}$$

For example, the condition (6) is satisfied if the noise of the gradient estimate for each coordinate is *unimodal and symmetric about its mean*. We use this assumption in our paper, but other assumptions (Safaryan & Richtárik, 2021) leading to (6) are valid as well.

---
**Algorithm 3** MajorityVote-SignSGD
---
**Input:** Starting point $x^0 \in \mathbb{R}^d$, number of iterations $T$, stepsizes $\{\gamma_k\}_{k=1}^T$, batchsizes $\{M_k\}_{k=1}^T$.
1: **for** $k = 1, \dots, T$ **do**
2:     Sample $\{\xi_i^k\}_{i=1}^{B_k}$ and compute gradient estimate
    $g^k = \sum_{i=1}^{M_k} \text{sign}(\nabla f(x^k, \xi_i^k))$;
3:     Set $x^{k+1} = x^k - \gamma_k \cdot \text{sign}(g^k)$;
4: **end for**
**Output:** uniformly random point from $\{x^1, \dots, x^T\}$.

---

**Theorem 2** (**HP complexity for** MajorityVote-SignSGD). *Consider lower-bounded $L$-smooth function $f$ (As. 1, 2) and gradient estimates corrupted by **unimodal and symmetric** HT noise (As. 3). Then Alg. 3 requires the sample complexity $N$ to achieve $\frac{1}{T}\sum_{k=1}^T \|\nabla f(x^k)\|_1 \le \varepsilon$ with probability at least $1 - \delta$ for:*

*arbitrary tuning:* $T, \gamma_k \equiv \frac{\gamma_0}{\sqrt{T}}, M_k \equiv \max\{1, M_0 T\}$:

$$N = O\left( \frac{M_0(\Delta_1/\gamma_0 + L_\delta d\gamma_0 + a_\kappa \|\vec{\sigma}\|_1/\sqrt{M_0})^4}{\varepsilon^4} \right), \tag{7}$$

*optimal tuning:* $T = O\left( \frac{\Delta_1 L_\delta d}{\varepsilon^2} \right), \gamma_k \equiv \sqrt{\frac{\Delta_1}{8L_\delta dT}}, M_k \equiv \max\left\{ 1, \left( 8a_\kappa \frac{\|\vec{\sigma}\|_1}{\varepsilon} \right)^2 \right\}$:

$$N = O\left( \frac{\Delta_1 L_\delta d}{\varepsilon^2} + \frac{\Delta_1 L_\delta d}{\varepsilon^2}\left( \frac{a_\kappa \|\vec{\sigma}\|_1}{\varepsilon} \right)^2 \right), \tag{8}$$

*where $\Delta_1 = f(x^1) - f^*, L_\delta = L\log 1/\delta, (a_\kappa)^\kappa := \frac{\kappa+1}{\kappa-1}$.*

The bound (8) matches minibatch-SignSGD bound (4) if $\kappa = 2$. The dependency on $\kappa$ is expressed in slowing degenerating multiplicative factor $\alpha_\kappa$ instead of $\varepsilon^{-\frac{\kappa}{\kappa-1}}$.

**Remark 1.** *In Appendix A, we provide a method built on top of* minibatch-SignSGD *algorithm and majority voting for the distributed setup with the fixed number of workers.*

## 2.5. SignSGD with momentum

Instead of variance reduction, one can use momentum technique with the same sample comlexity.

---

**Algorithm 4** M-SignSGD

---

**Input:** Starting point $x^1 \in \mathbb{R}^d$, number of iterations $K$, stepsizes $\{\gamma_k\}_{k=1}^T$, momentums $\{\beta_k\}_{k=1}^T$.
1: **for** $k = 1, \dots, T$ **do**
2:   Sample $\xi^k$ and compute estimate $g^k = \nabla f(x^k, \xi^k)$;
3:   Compute $m^k = \beta_k m^{k-1} + (1 - \beta_k) g^k$;
4:   Set $x^{k+1} = x^k - \gamma_k \cdot \text{sign}(m^k)$;
5: **end for**
**Output:** uniformly random point from $\{x^1, \dots, x^T\}$.

---

**Theorem 3** (**Complexity for** M-SignSGD **in expectation**). *Consider lower-bounded L-smooth function $f$ (As. 1, 2) and HT gradient estimates (As. 3). Then Alg. 4 requires $T$ iterations to achieve $\frac{1}{T} \sum_{k=1}^T \mathbb{E}\left[\|\nabla f(x^k)\|_1\right] \le \varepsilon$ for:*

**arbitrary tuning:** $T, \gamma_k \equiv \gamma_0 T^{-\frac{3}{4}}, \beta_k \equiv 1 - 1/\sqrt{T}$:

$$T = O\left(\frac{(\Delta_1/\gamma_0 + Ld\gamma_0)^4}{\varepsilon^4} + \left(\frac{\sqrt{d}\|\vec{\sigma}\|_\kappa}{\varepsilon}\right)^{\frac{2\kappa}{\kappa-1}}\right),$$

**optimal tuning:** $\gamma_k \equiv \sqrt{\frac{\Delta_1(1-\beta_k)}{4LdT}}, \beta_k \equiv 1 - \min\left\{1, \frac{1}{\|\vec{\sigma}\|_\kappa^2} \cdot \left(\frac{\Delta_1 L}{T}\right)^{\frac{\kappa}{3\kappa-2}}\right\}$:

$$T = O\left(\frac{\Delta_1 Ld}{\varepsilon^2} + \frac{\Delta_1 Ld}{\varepsilon^2}\left(\frac{\sqrt{d}\|\vec{\sigma}\|_\kappa}{\varepsilon}\right)^{\frac{\kappa}{\kappa-1}}\right), \quad (9)$$

*where $\Delta_1 = f(x^1) - f^*$.*

In comparison with (4) for minibatch-SignSGD, in expectation bound (9) has a larger $\sqrt{d}\|\vec{\sigma}\|_\kappa$ factor instead of $\|\vec{\sigma}\|_1$, but they are still close due to the norm relation (10).

## 2.6. Discussion and comparison with related works

**Optimality.** First, we compare theoretical complexities. In (Zhang et al., 2020b), the authors provided lower bound in expectation for the sample complexity $\Omega\left(\frac{\Delta_1 L}{\varepsilon^2} + \frac{\Delta_1 L}{\varepsilon^2}\left(\frac{\|\vec{\sigma}\|_\kappa}{\varepsilon}\right)^{\frac{\kappa}{\kappa-1}}\right)$ w.r.t. the $\ell_2$-norm for non-convex functions. Our bounds are the first-known

bounds with HP for any HT noise: minibatch-SignSGD attains bound $O\left(\frac{\Delta_1 Ld \log 1/\delta}{\varepsilon^2} + \frac{\Delta_1 Ld \log 1/\delta}{\varepsilon^2}\left(\frac{\|\vec{\sigma}\|_1}{\varepsilon}\right)^{\frac{\kappa}{\kappa-1}}\right)$ w.r.t. the $\ell_1$-norm with linear $\log 1/\delta$ dependency. However, there are $d$ factors and $\|\vec{\sigma}\|_1$ instead of smaller $\|\vec{\sigma}\|_\kappa$. Since

$$\|x\|_2 \le \|x\|_1 \le \sqrt{d}\|x\|_2, \forall x \in \mathbb{R}^d, \quad (10)$$

in order to achieve $\varepsilon$ accuracy in the $\ell_2$-norm, accuracy $\varepsilon'$ in the $\ell_1$-norm has to be $\varepsilon' = \varepsilon \cdot \sqrt{d}$. Thus, the total number of samples and the optimality remain. A good analysis of the relation between convergence w.r.t. the different norms is given in (Bernstein et al., 2018a). For M-SignSGD, (9) exactly matches the optimal bound with the same remarks.

**Clipping.** According to the HP analysis of ClipSGD from (Sadiev et al., 2023, Theorem 3.1), it achieves before mentioned lower bound with extra $\log 1/\varepsilon$ and $\log 1/\delta$ factors. Moreover, the constants concealed behind $O$ notation have $10^3$ magnitudes. To compare, sign-based methods have smaller constants without extra accuracy factors. From the practical point of view, clipping levels depend on the iteration number and affect the final accuracy without a good tuning (Sadiev et al., 2023, Theorem 3.1). The sign-based methods work well with constant, arbitrary parameters.

**Normalized SGD.** In (Hübler et al., 2024), the authors analyze HP convergence of normalization-based methods under HT noise. These methods use normalization $g^k/\|g^k\|_2$ instead of $\text{sign}(g^k)$. Namely, for non-convex functions, minibatch-NSGD with the same optimal batchsizes has sample complexity w.r.t. to the $\ell_2$-norm:

$$O\left(\frac{\Delta_1 L_\delta}{\varepsilon^2} + \frac{\Delta_1 L_\delta}{\varepsilon^2}\left(\frac{\|\vec{\sigma}\|_\kappa}{\varepsilon}\right)^{\frac{\kappa}{\kappa-1}}\right). \quad (11)$$

The only difference of (11) from (4) is the absence of $d$ factors, which can be explained by different norm for required accuracy. The same comparison is valid for the momentum methods. From the practical point of view, sign-based methods can be applied to distributed optimization (Appendix A) where normalization does not fit. Besides, one can use majority voting as a more powerful alternative to batching.

**Symmetric HT noise.** For MajorityVote-SignSGD, the bounds (7), (8) match optimal bounds in expectation for first-order non-convex methods with *BV noise* (Arjevani et al., 2023) with mild $\log 1/\delta$ dependency. In (Armacki et al., 2023; 2024), the authors considered only symmetric noise and proved bounds *arbitrary close* to $O(\varepsilon^{-4})$ in online paradigm. On the contrary, our bounds are tight.

## 3. High probability bounds for comparison oracle under heavy-tailed noise

In this section, we switch from the first-order optimization and gradient oracle to the zeroth-order optimization and

an oracle which can only compare two corrupted function values at two different points. For non-convex functions, we propose our novel MajorityVote-CompSGD method and prove its HP bounds under the symmetric HT noise. All proofs are located in Appendix C.

### 3.1. Comparison oracle

For any two points $x, y \in \mathbb{R}^D$, the **stochastic comparison oracle** $\phi(x, y, \xi = \{\xi_x, \xi_y\})$ determines which noisy function value, $f(x, \xi_x)$ or $f(y, \xi_y)$, is larger (the realizations $\xi_x$ and $\xi_y$ may be dependent):

$$\phi(x, y, \xi) = \text{sign}(f(x, \xi_x) - f(y, \xi_y)).$$

This oracle concept is natural for describing human decision-making (Lobanov et al., 2024a). Given a choice between two options, it is usually much easier to choose which option is better rather than estimate quantative difference. The stochasticity $\xi$ describes the variety of division-makers and their random states. For example, this oracle is extensively used in Reinforcement Learning (RL) and training Large Language Models via RL with human feedback (Ouyang et al., 2022; Wang et al., 2023; Tang et al., 2023).

### 3.2. Related works

The most common instance of methods using comparisons is Stochastic Three Points (STP) (Bergou et al., 2020; Gorbunov et al., 2022). It takes a random direction and goes along it where the function value is smaller. Initially STP was analyzed for non-convex functions without any noise. In (Boucherouite et al., 2024), the authors worked with sum-type functions and stochastic minibatches. They proved convergence of STP in expectation under BV noise, but with the obligatory condition on huge batchsizes.

In (Saha et al., 2021), the authors considered a noisy comparison oracle where noise was introduced as a fixed probability of receiving a wrong sign during comparison. They restated STP via sign operator and at each iteration repeated Bernoulli trials with comparisons to ensure the sign correctness with high confidence. The authors obtained HP bounds, but only for convex and strongly-convex functions.

There are other approaches to incorporate comparison oracle. In (Lobanov et al., 2024a), the authors used Coordinate Gradient Descent (CGD) with the search of the steepest stepsizes via golden ration method. In the deterministic case with adversarial (non-stochastic) noise, this approach achieved better parameter dependencies and practical results. Especially, for strongly convex functions, for which the authors used accelerated CGD. The authors also proposed an algorithm for stochastic oracle and prove its asymptotic convergence. In (Tang et al., 2023), comparison oracle was used to build a ranking-based gradient estimate over random directions which was then plugged into GD.

### 3.3. Assumptions

**Heavy-tailed noise.** We consider the following corrupting heavy-tailed noises induced by variable $\xi$:

**Assumption 4** (Heavy-tailed noise in function estimates). *The function estimate $f(x, \xi)$ is unbiased and has bounded $\kappa$-th moment $\kappa \in (1, 2]$ with $\sigma > 0$, i.e.,*

*1)* $\mathbb{E}_\xi[f(x, \xi)] = f(x), \quad \forall x \in \mathbb{R}^d,$

*2)* $\mathbb{E}_\xi[|f(x, \xi) - f(x)|^\kappa] \leq \sigma^\kappa, \quad \forall x \in \mathbb{R}^d.$

*For $\kappa = 2$, the noise is called bounded variance.*

For example, the estimate $f(x, \xi)$ can be corrupted at each point by independent heavy-tailed noise $\xi$ with bounded $\kappa$-th moment: $f(x, \xi) := f(x) + \xi$.

Another example of such estimate is when we optimize a sum-type function $f(x) = \frac{1}{K} \sum_{i=1}^K f_i(x)$, and $\xi$ denotes a random batch $I$ of fixed size $|I|$ from $\{1, \ldots, K\}$, i.e., $f(x, \xi) = \frac{1}{|I|} \sum_{i \in I} f_i(x)$. For two points, oracle gives the same $\xi$ realization (batch). This estimate satisfies discrete BV noise assumption (Boucherouite et al., 2024).

**Random directions.** We use the following assumption on the random directions' distribution $\mathcal{D}$:

**Assumption 5** (Random directions). *The distribution $\mathcal{D}$ on $\mathbb{R}^d$ has the following properties:*

*1) There exist a norm $\|\cdot\|_p, p \in [1, 2]$ and a constant $\alpha_p > 0$, such that for all $g \in \mathbb{R}^d$:*

$$\mathbb{E}_{\mathbf{e} \in \mathcal{D}} |\langle g, \mathbf{e} \rangle| \geq \alpha_p \|g\|_p.$$

*2) For all $\mathbf{e} \in \mathcal{D}$, the norms $\|\mathbf{e}\|_2 \leq 1, \|\mathbf{e}\|_q \leq 1, \frac{1}{p} + \frac{1}{q} = 1$.*

We use the following instances of $\mathcal{D}$ with explicit constants and norms (Bergou et al., 2020, Lemma 3.4):

**1)** Uniform distribution on Euclidean sphere $S_2^d := \{\mathbf{e} | \quad \|\mathbf{e}\|_2 = 1\}, p = 2, \alpha_p = \frac{1}{\sqrt{2\pi d}}$.

**2)** Uniform distribution on standard basic vectors $\{e_1, \ldots, e_d\}, p = 1, \alpha_p = \frac{1}{d}$.

### 3.4. CompSignSGD and its convergence properties

In (Lobanov et al., 2024a), the authors propose a nameless procedure for stochastic oracle:

$$x^{k+1} = x^k - \gamma_k \cdot \text{sign}(f(x^k + \gamma_k \mathbf{e}^k, \xi_+) - f(x^k - \gamma_k \mathbf{e}^k, \xi_-)) \cdot \mathbf{e}^k.$$

If value $f(x^k - \gamma_k \mathbf{e}^k, \xi_-)$ is smaller than $f(x^k + \gamma_k \mathbf{e}^k, \xi_+)$, then sign equals to 1 and $x^{k+1} = x^k - \gamma_k \mathbf{e}^k$. Otherwise, point $x^{k+1} = x^k - \gamma_k \mathbf{e}^k$ is chosen. We name it CompSGD (Alg. 5) and prove the following convergence Lemma.

**Lemma 2** (CompSGD **Convergence Lemma**). *Consider lower-bounded $L$-smooth function $f$ (As. 1, 2), random*

**Algorithm 5** CompSGD

---

**Input:** Starting point $x^1 \in \mathbb{R}^d$, number of iterations $T$, stepsizes $\{\gamma_k\}_{k=1}^T$;
  1: **for** $k = 1, \dots, T$ **do**
  2:    Sample direction $\mathbf{e}^k$ and $\xi^k$ ;
  3:    $\phi^k := \text{sign}\left[ f(x^k + \gamma_k \mathbf{e}^k, \xi_+^k) - f(x^k - \gamma_k \mathbf{e}^k, \xi_-^k) \right]$;
  4:    Set $x^{k+1} = x^k - \gamma_k \cdot \phi^k \cdot \mathbf{e}^k$;
  5: **end for**
**Output:** uniformly random point from $\{x^1, \dots, x^T\}$ ;

---

directions with $\alpha_p$ (As. 5) and HT function estimates (As. 4). Then Alg. 5 after $T$ iteration with constant stepsizes $\gamma_k \equiv \gamma$ achieves with probability at least $1 - \delta$ starting with $\Delta_1 = f(x^1) - f^*$:

$$\frac{1}{T} \sum_{k=1}^T \|\nabla f(x^k)\|_p \leq \frac{2\Delta_1}{T\alpha_p\gamma} + \frac{12d^{\frac{1}{p}}\|\nabla f(x^0)\|_2}{T\sqrt{d}\alpha_p} \log(1/\delta)$$
$$+ \; 24\frac{Ld^{\frac{1}{p}}\gamma}{\sqrt{d}\alpha_p} \log(1/\delta) + \frac{8\sigma}{\alpha_p\gamma}.$$

As a consequence, in order to achieve accuracy $\varepsilon$, the noise $\sigma$ must not exceed $\sigma \sim \alpha_p\varepsilon^2$.

### 3.5. Our MajorityVote-CompSGD

At this point, we propose our novel MajorityVote-CompSGD which can reduce noise via the majority voting over comparison signs:

$$x^{k+1} = x^k - \gamma_k \text{sign}\left[ \sum_{i=1}^M \phi(x^k + \gamma_k \mathbf{e}^k, x^k - \gamma_k \mathbf{e}^k, \xi_i^k) \right]\mathbf{e}^k.$$

---

**Algorithm 6** MajorityVote-CompSignSGD

---

**Input:** Starting point $x^1 \in \mathbb{R}^d$, number of iterations $T$, stepsizes $\{\gamma_k\}_{k=1}^T$, batchsizes $\{M_k\}_{k=1}^T$.
  1: **for** $k = 1, \dots, T$ **do**
  2:    Sample direction $\mathbf{e}^k$ and $\{\xi_i^k\}_{i=1}^{M_k}$;
  3:    $\phi_i^k = \text{sign}\left[ f(x^k + \gamma_k\mathbf{e}^k, \xi_{i,+}^k) - f(x^k - \gamma_k\mathbf{e}^k, \xi_{i,-}^k) \right]$ ;
  4:    Set $x^{k+1} = x^k - \gamma_k \cdot \text{sign}\left( \sum_{i=1}^{M_k} \phi_i^k \right) \cdot \mathbf{e}^k$;
  5: **end for**
**Output:** uniformly random point from $\{x^1, \dots, x^T\}$ .

---

Similar to MajorityVote-SignSGD, we require additional assumption of unimodality and symmetry of HT noise $f(x, \xi) - f(x), \forall x \in \mathbb{R}^d$.

**Theorem 4** (**HP complexity for** MajorityVote–CompSGD)**.** *Consider lower-bounded L-smooth function f (As. 1, 2), random directions with $\alpha_p$ (As. 5) and HT **unimodal and symmetric** function estimates (As.*

4*). Then Alg. 6 requires comparison number $N$ to achieve* $\frac{1}{T} \sum_{k=1}^T \|\nabla f(x_k)\|_p \leq \varepsilon$ *with probability at least $1 - \delta$ for:*

**arbitrary tuning:** $T, \gamma_k \equiv \gamma_0/\sqrt{T}, M_k \equiv \max\{1, M_0T^2\}$:

$$N = M_0T^3 = O\left( \frac{M_0 \cdot ((\Delta_1 + a_\kappa\sigma/\sqrt{M_0})/\gamma_0 + L_{\delta,p}\gamma_0)^6}{(\alpha_p\varepsilon)^6} \right),$$

**optimal tuning:** $T = O\left( \frac{\Delta_1 L_{\delta,p}}{\alpha_p^2\varepsilon^2} \right), \gamma_k \equiv \sqrt{\frac{\Delta_1}{12L_{\delta,p}T}}$ *and* $M_k \equiv \max\left\{ 1, \left( \frac{32a_\kappa\sigma}{\alpha_p\varepsilon\gamma} \right)^2 \right\}$:

$$N = O\left( \frac{\Delta_1 L_{\delta,p}}{\alpha_p^2\varepsilon^2} + \frac{\Delta_1 L_{\delta,p}}{\alpha_p^2\varepsilon^2}\left( \frac{a_\kappa\sigma L_{\delta,p}}{\alpha_p^2\varepsilon^2} \right)^2 \right), \quad (12)$$

*where* $\Delta_1 = f(x^1) - f^*, L_{\delta,p} = L\log(\frac{1}{\delta})d^{\frac{1}{p} - \frac{1}{2}}, (a_\kappa)^\kappa = \frac{\kappa+1}{\kappa-1}$.

**Remark 2** (CompSGD **with function batching**)**.** *If one can directly compare the batched function values at two points, e.g. with the sum-type objective functions, then batch averaging can be applied under any HT noise. Similar to* minibatch-SignSGD, *we substitute the step 3 in Alg. 6 with* $g^k = \text{sign}\left[ \sum_{i=1}^{B_k} f(x^k + \gamma_k\mathbf{e}^k, \xi_{i,+}^k) - \sum_{i=1}^{B_k} f(x^k - \gamma_k\mathbf{e}^k, \xi_{i,-}^k) \right]$ *and build new* minibatch-CompSGD *method (See Appendix B). It achieves the number of function calls $N$ with HP with the optimal parameters* $T = O\left( \frac{\Delta_1 L_{\delta,p}}{\alpha_p^2\varepsilon^2} \right), \gamma_k \equiv \sqrt{\frac{\Delta_1}{TL_{\delta,p}}}$ *and* $B_k \equiv \max\left\{ 1, \left( \frac{\sigma T}{\Delta_1} \right)^{\frac{\kappa}{\kappa-1}} \right\}$:

$$N = O\left( \frac{\Delta_1 L_{\delta,p}}{\alpha_p^2\varepsilon^2} + \frac{\Delta_1 L_{\delta,p}}{\alpha_p^2\varepsilon^2} \cdot \left( \frac{\sigma L_{\delta,p}}{\alpha_p^2\varepsilon^2} \right)^{\frac{\kappa}{\kappa-1}} \right). \quad (13)$$

### 3.6. Discussion and comparison with related works

**Optimality.** When $\mathcal{D}$ is a Euclidean sphere, the first term $\tilde{O}(dL/\varepsilon^2)$ from (12), (13) matches the bounds for noiseless methods from previous works (Bergou et al., 2020; Tang et al., 2023; Lobanov et al., 2024a) and the optimal bound for the deterministic zeroth-order optimization (Nemirovskij & Yudin, 1983). Moreover, our HP threshold on noise $\sigma \sim \varepsilon^2/\sqrt{d}$ is the same as the threshold for adversarial noise from (Lobanov et al., 2024a) or for batched variance from (Boucherouite et al., 2024). This threshold is optimal w.r.t. $\varepsilon$ and $d$ (Lobanov, 2023). Hence, our bounds are tight.

**Comparison.** Although CompSGD was proposed in (Lobanov et al., 2024a), the authors proved only its *asymptotic convergence with parameters depending on the solution*. We prove its explicit formulas with HP (Lemma 2) and propose novel modification (Alg. 6) which converges non-asymptotically (Th. 4).

The noisy comparison oracle from (Saha et al., 2021) is similar to ours. The authors used a non-trivial assumption:

$$\mathbb{P}_\xi \left[ \phi(x, y, \xi) \neq \text{sign}(f(x) - f(y)) \right] \leq 1/2 - \nu, \forall x, y \in \mathbb{R}^d, \tag{14}$$

for some constant $\nu \in (0, 1/2)$. First, all results from (Saha et al., 2021) are proved for the convex functions, and we prove it for the non-convex ones. Next, we highlight that *our Assumption 4 is much weaker and general*, since (14) can fail even under BV noise. In proofs, we show that

$$\mathbb{P}_\xi \left[ \phi(x, y, \xi) \neq \text{sign}(f(x) - f(y)) \right] \leq \sigma / |f(x) - f(y)|.$$

Thus, in the vicinity of the stationary point where function changes are small or under the large $\sigma$, (14) can not hold.

## 4. Experiments

In this section, we present experimental results for both first-order and zero-order methods described in Sections 2 and 3, respectively. To demonstrate the superiority of sign-based methods for problems with heavy-tailed noise, we focus on language model training tasks. This choice is motivated by two factors: first, these tasks are known to exhibit heavy-tailed noise characteristics (Zhang et al., 2020c), and second, they represent an important real-world application domain.

### 4.1. M-SignSGD on LLaMA pre-training

First, we evaluate the performance of M-SignSGD (Algorithm 4) on the language model pre-training task. We adopt the established experimental setup from (Lialin et al., 2023), training a 130M parameter LLaMA-like model (Touvron et al., 2023) on the Colossal Clean Crawled Corpus (C4) dataset (Raffel et al., 2020). The C4 dataset represents a comprehensive, sanitized version of Common Crawl's web corpus, specifically designed for pre-training language models and word representations.

For our comparison, we focus on two key techniques for handling heavy-tailed noise: gradient clipping with momentum and gradient normalization with momentum. As representative methods, we choose M-ClippedSGD (Zhang et al., 2020a) and M-NSGD (Cutkosky & Mehta, 2020), respectively. We also compare to AdamW (Loshchilov, 2017), as a de-facto method for first-order optimization algorithm for deep learning. To ensure a fair comparison, we conduct an extensive grid search over key hyperparameters, including learning rate, weight decay, and clipping level. Detailed information about the final hyperparameter values and complete experimental setup is provided in Appendix D.1.

Table 1 presents final validation perplexity for each method. M-SignSGD demonstrates superior performance over the baselines, aligning with our theoretical results.

Table 1: Perplexity of LLaMA-130M model pre-trained on C4 for 100k steps. Lower is better.

| Method | Perplexity ↓ |
|---|---|
| M-SignSGD | **18.37** |
| M-NSGD | 19.28 |
| M-ClippedSGD | 18.95 |
| AdamW | 18.67 |

Table 2: Accuracy of RoBERTa-large (350M parameters) fine-tuned on different tasks. Higher is better.

| Method | SST-2 | MNLI | TREC |
|---|---|---|---|
| CompSGD | **91.9** | **63.8** | **77.2** |
| MeZO | 91.7 | 58.7 | 76.9 |
| Zero-shot | 79.0 | 48.8 | 32.0 |

### 4.2. CompSGD on RoBERTa fine-tuning

Second, we consider zeroth-order setting. Following MeZO (Malladi et al., 2023a), we evaluate our method on classification fine-tuning tasks, specifically SST-2 (Socher et al., 2013), MNLI (Williams et al., 2017), TREC (Voorhees & Tice, 2000), on the RoBERTa-large model (Liu et al., 2019b). We employ the established few-shot prediction setting (Malladi et al., 2023b; Gao et al., 2020a). See details in Appendix D.2.

We compare CompSGD Algorithm 5 against pre-trained model without fine-tuning (Zero-shot) and original MeZO version. As demonstrated in Table 2, the sign-based method again outperforms its non-sign counterpart.

### 4.3. CompSGD for accuracy maximization

Third, we simulate the zeroth-order environment with comparison oracles as follows. We take the prediction accuracy of the linear model on the training dataset as the target:

$$f(x) = \left(1 - \text{Acc}\left(\mathbf{y}_{\text{train}}, \text{sign}\left(\frac{2}{1+\exp(-\mathbf{X}_{\text{train}}x)}\right) - 1\right)\right).$$

As training data, we consider classification tasks from LibSVM (Chang & Lin, 2011): mushrooms, phishing, a6a. In Figure 1, we give the dynamics of accuracy on the test sample for our method and for another method working with the comparison oracle OrderRCD (Lobanov et al., 2024b). Here, we also outperformed the competitor.

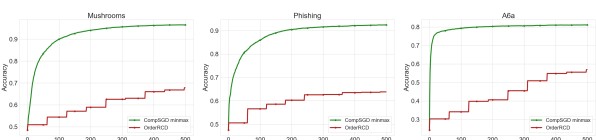

Figure 1: Performance of zeroth-order methods with comparison oracle.

## Impact Statements

This paper presents work whose goal is to advance the field of Machine Learning. There are many potential societal consequences of our work, none which we feel must be specifically highlighted here.

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

## A. SignSGD **for the distributed optimization**

Consider the distributed optimization with one server and $M$ workers, each of which calculates its own gradient estimate. The server receives all estimates, aggregates them and sends back the updated solution to the workers. The sign-based methods are so effective in terms of communication (Bernstein et al., 2018b; Jin et al., 2020), since sending a sign vector costs only $O(d)$ operations. We use the aggregation based on the majority voting.

---

**Algorithm 7** Distributed-MajorityVote-SignSGD

---

**Input:** Starting point $x^1 \in \mathbb{R}^d$, number of iterations $T$, stepsizes $\{\gamma_k\}_{k=1}^T$, batchsizes $\{B_k\}_{k=1}^T$.
 1: **for** $k = 1, \ldots, T$ **do**
 2:   Sample $\{\xi_i^{k,j}\}_{i=1}^{B_k}$ and compute gradient estimate $g^{k,j} = \sum_{i=1}^{B_k} \nabla f(x^k, \xi_i^{k,j})/B_k$ for each worker $j \in \overline{1, M}$;
 3:   Send signs $\mathrm{sign}(g^{k,j})$ to server for each worker $j \in \overline{1, M}$;
 4:   Compute on server $g^k = \mathrm{sign}\left(\sum_{j=1}^M \mathrm{sign}(g^{k,j})\right)$;
 5:   Send point $x^{k+1} = x^k - \gamma_k \cdot g^k$ to each worker;
 6: **end for**
**Output:** uniformly random point from $\{x^1, \ldots, x^T\}$.

---

**Theorem 5** (**HP complexity for** Distributed-MajorityVote-SignSGD). *Consider lower-bounded $L$-smooth function $f$ (As. 1, 2) and gradient estimates corrupted by **unimodal and symmetric** HT noise (As. 3). Then Alg. 7 with $M$ workers requires the sample complexity $N_M$ per worker to achieve $\frac{1}{T}\sum_{k=1}^T \|\nabla f(x^k)\|_1 \leq \varepsilon$ with probability at least $1 - \delta$ for:*

*arbitrary tuning:* $T, \gamma_k \equiv \frac{\gamma_0}{\sqrt{T}}, B_k \equiv \max\{1, B_0 T\}$:

$$N_M = O\left(\frac{B_0(\Delta_1/\gamma_0 + L_\delta d\gamma_0)^4}{\varepsilon^4} + \frac{1}{B_0}\left(\frac{a_\kappa\|\vec{\sigma}\|_1}{\sqrt{M}\varepsilon}\right)^{\frac{2\kappa}{\kappa-1}}\right),$$

*optimal tuning:* $T = O\left(\frac{\Delta_1 L_\delta d}{\varepsilon^2}\right), \gamma_k \equiv \sqrt{\frac{\Delta_1}{8dL_\delta T}}, B_k \equiv \max\left\{1, \left(\frac{16a_\kappa\|\vec{\sigma}\|_1}{\sqrt{M}\varepsilon}\right)^{\frac{\kappa}{\kappa-1}}\right\}$:

$$N_M = O\left(\frac{\Delta_1 L_\delta d}{\varepsilon^2} + \frac{\Delta_1 L_\delta d}{\varepsilon^2}\left(\frac{a_\kappa\|\vec{\sigma}\|_1}{\sqrt{M}\varepsilon}\right)^{\frac{\kappa}{\kappa-1}}\right),$$

*where $\Delta_1 = f(x^1) - f^*, L_\delta = L\log 1/\delta, (a_\kappa)^\kappa := \left(\frac{\kappa+1}{\kappa-1}\right).$*

The proof of Theorem 5 is located in Appendix C.5.

**Remark 3** (SignSGD **with median clipping**). *For the symmetric HT noise with a mild condition on probability density function, there exists a complement to the batch averaging, namely, coordinate-wise median operator (Puchkin et al., 2024). For all $\kappa \in (1, 2]$, it requires only $9$ samples to build an unbiased BV gradient estimate. Then it can be combined with* minibatch-SignSGD *as if $\kappa = 2$. In this case, the $\kappa$ dependency from Theorems 2 and 5 can be completely removed.*

## B. CompSGD **with function batching**

If one can batch function values at two points before its direct comparison (e.g. with sum-type objective function), then CompSGD combined with the batching achieves the following bounds.

**Theorem 6** (**HP complexity for** minibatch-CompSGD). *Consider lower-bounded $L$-smooth function $f$ (As. 1, 2), random directions with $\alpha_p$ (As. 5) and HT function estimates with $\sigma, \kappa \in (1, 2]$ (As. 4). Then Alg. 8 requires $N$ function calls to achieve $\frac{1}{T}\sum_{k=1}^T \|\nabla f(x_k)\|_p \leq \varepsilon$ with probability at least $1 - \delta$ for:*

*optimal tuning:* $T = O\left(\frac{\Delta_1 L_{\delta,p}}{\alpha_p^2\varepsilon^2}\right), \gamma_k \equiv\equiv \sqrt{\frac{\Delta_1}{TL_{\delta,p}}}$ *and* $B_k \equiv \max\left\{1, \left(\frac{\sigma T}{\Delta_1}\right)^{\frac{\kappa}{\kappa-1}}\right\}$:

$$N = O\left(\frac{\Delta_1 L_{\delta,p}}{\alpha_p^2\varepsilon^2} + \frac{\Delta_1 L_{\delta,p}}{\alpha_p^2\varepsilon^2} \cdot \left(\frac{\sigma L_{\delta,p}}{\alpha_p^2\varepsilon^2}\right)^{\frac{\kappa}{\kappa-1}}\right), \tag{15}$$

---

**Algorithm 8** minibatch-CompSGD

---

**Input:** Starting point $x^1 \in \mathbb{R}^d$, number of iterations $T$, stepsizes $\{\gamma_k\}_{k=1}^T$, batchsizes $\{B_k\}_{k=1}^T$.

  1: **for** $k = 1, \ldots, T$ **do**

  2:      Sample direction $\mathbf{e}^k$ and $\{\xi_i^k\}_{i=1}^{B_k}$;

  3:      Compare $g^k = \text{sign}\left( \sum\limits_{i=1}^{B_k} f(x^k + \gamma_k \mathbf{e}^k, \xi_i^k) - \sum\limits_{i=1}^{B_k} f(x^k - \gamma_k \mathbf{e}^k, \xi_i^k) \right)$;

  4:      Set $x^{k+1} = x^k - \gamma_k \cdot g^k \cdot \mathbf{e}^k$;

  5: **end for**

**Output:** uniformly random point from $\{x^1, \ldots, x^T\}$ .

---

*where* $\Delta_1 = f(x^1) - f^*$, $L_{\delta,p} = d^{\frac{1}{p} - \frac{1}{2}} L \log 1/\delta$.

The proof of Theorem 6 is located in Appendix C.7. For two considered distributions $\mathcal{D}$, the Euclidean sphere and the standard basis, we estimate the value $\alpha_p \varepsilon$. Euclidean sphere's value $\alpha_p$ is large by a factor $\sqrt{d}$. However, the sphere itself induces the $\ell_2$-norm in the final estimate (13). Due to the inequality

$$\|\nabla f(x)\|_2 \le \|\nabla f(x)\|_1 \le \sqrt{d}\|\nabla f(x)\|_2,$$

in order to achieve the same $\varepsilon$ bound for the $\ell_2$-norm with the standard basis, the actual $\ell_1$-accuracy $\varepsilon'$ must $\varepsilon' = \varepsilon * \sqrt{d}$. Hence, the value $\alpha_p \varepsilon$ for both setups is the same.

# C. Proofs

## C.1. Technical lemmas and propositions

We use the following facts from the linear algebra and convex analysis (Boyd, 2004):

**Proposition 1** (Smoothness inequality). *For L-smooth function $f$ (As. 2), the following inequality holds true*

$$f(y) - f(x) - \langle \nabla f(x), y - x \rangle \le \frac{L}{2}\|x - y\|_2^2, \quad \forall x, y \in \mathbb{R}^d. \tag{16}$$

**Proposition 2** (Norm Relation). *For two norms $\ell_p$ and $\ell_q$ with $1 \le p \le q \le 2$, the following relation holds true:*

$$\|x\|_q \le \|x\|_p \le d^{\frac{1}{p} - \frac{1}{q}}\|x\|_q, \quad \forall x \in \mathbb{R}^d. \tag{17}$$

**Proposition 3** (Jensen's Inequality). *For scalar random variable $\xi$ with bounded $\kappa$-th moment $\kappa \in (1, 2]$, the following inequality holds true:*

$$\mathbb{E}[|\xi|] \le (\mathbb{E}[|\xi|^\kappa])^{\frac{1}{\kappa}}. \tag{18}$$

**Proposition 4** (Markov's Inequality). *For scalar random variable $\xi$ with bounded first moment, the following inequality holds true for any $a > 0$:*

$$\mathbb{P}(|\xi - \mathbb{E}[\xi]]| \ge a) \le \frac{\mathbb{E}[|\xi|]}{a}. \tag{19}$$

To prove the HP bounds with the logarithmic dependency, we use the following measure concentration result (see, for example, (Li & Orabona, 2020, Lemma 1).

**Lemma 3** (Measure Concentration Lemma). *Let $\{D_k\}_{k=1}^T$ be a martingale difference sequence (MDS), i.e., $\mathbb{E}[D_k | D_{k-1}, \ldots, D_1] = 0$ for all $k \in \overline{1, T}$. Furthermore, for each $k \in \overline{1, T}$, there exists positive $\sigma_k \in \mathbb{R}$, s.t. $\mathbb{E}\left[\exp\left(\frac{D_k^2}{\sigma_k^2}\right) | k\right] \le e$. Then the following probability bound holds true:*

$$\forall \lambda > 0, \delta \in (0, 1): \quad \mathbb{P}\left( \sum_{k=1}^T D_k \le \frac{3}{4}\lambda \sum_{k=1}^T \sigma_k^2 + \frac{1}{\lambda}\log(1/\delta) \right) \ge 1 - \delta. \tag{20}$$

To control error reduction during batching, we use the following batching lemma for HT variables. Its modern proof for $d = 1$ was proposed in (Cherapanamjeri et al., 2022, Lemma 4.2) and then generalized for the multidimensional case in (Kornilov et al., 2024; Hübler et al., 2024).

**Lemma 4** (HT Batching Lemma). *Let* $\kappa \in (1, 2]$*, and* $X_1, \ldots, X_B \in \mathbb{R}^d$ *be a martingale difference sequence (MDS), i.e.,* $\mathbb{E}[X_i|X_{i-1}, \ldots, X_1] = 0$ *for all* $i \in \overline{1, B}$*. If all variables* $X_i$ *have bounded* $\kappa-$*th moment, i.e.,* $\mathbb{E}[\|X_i\|_2^\kappa] < +\infty$*, then the following bound holds true*

$$\mathbb{E}\left[\left\|\frac{1}{B}\sum_{i=1}^B X_i\right\|_2^\kappa\right] \leq \frac{2}{B^\kappa}\sum_{i=1}^B \mathbb{E}[\|X_i\|_2^\kappa]. \tag{21}$$

We need the following lemma about changes after one update step of sign-based methods from (Sun et al., 2023, Lemma 1).

**Lemma 5** (Sign Update Step Lemma). *Let* $x, m \in \mathbb{R}^d$ *be arbitrary vectors,* $A = diag(a_1, \ldots, a_d)$ *be diagonal matrix and* $f$ *be L-smooth function (As. 2). Then for the update step*

$$x' = x - \gamma \cdot A \cdot \text{sign}(m)$$

*with* $\epsilon := m - \nabla f(x)$*, the following inequality holds true*

$$f(x') - f(x) \leq -\gamma\|A\nabla f(x)\|_1 + 2\gamma\|A\|_F\|\epsilon\|_2 + \frac{L\gamma^2\|A\|_F^2}{2}. \tag{22}$$

### C.2. Proof of SignSGD General Convergence Lemma 1

For beginning, we prove general lemma about SignSGD convergence with HT unbiased gradient estimates $g^k$ with $\vec{\sigma}, \kappa \in (1, 2)$. This proof considerably relies on proof techniques for NSGD from (Hübler et al., 2024).

*Proof.* Consider the $k$-th step of SignSGD. We use smoothness of function $f$ (Lemma 1) to estimate:

$$\begin{aligned} f(x^{k+1}) - f(x^k) &\leq& \langle \nabla f(x^k), x^{k+1} - x^k\rangle + \frac{L}{2}\|x^{k+1} - x^k\|_2^2 \\ &=& -\gamma_k\langle\nabla f(x^k), \text{sign}(g^k)\rangle + \frac{L\|\text{sign}(g^k)\|_2^2}{2}\gamma_k^2 \\ &=& -\gamma_k\frac{\langle\nabla f(x^k), \text{sign}(g^k)\rangle}{\|\nabla f(x^k)\|_1}\cdot\|\nabla f(x^k)\|_1 + \frac{Ld}{2}\gamma_k^2. \end{aligned}$$

Consequently, after summing all $T$ steps, we obtain:

$$\sum_{k=1}^T \gamma_k\frac{\langle\nabla f(x^k), \text{sign}(g^k)\rangle}{\|\nabla f(x^k)\|_1}\cdot\|\nabla f(x^k)\|_1 \leq \underbrace{f(x^1) - f(x^*)}_{=\Delta_1} + \frac{Ld}{2}\sum_{k=1}^T\gamma_k^2. \tag{23}$$

We introduce the following terms $\phi_k := \frac{\langle\nabla f(x^k), \text{sign}(g^k)\rangle}{\|\nabla f(x^k)\|_1} \in [-1, 1], \psi_k := \mathbb{E}[\phi_k|x^k]$ and $D_k := -\gamma_k(\phi_k - \psi_k)\|\nabla f(x^k)\|_1$. We note that $D_k$ is a martingale difference sequence ($\mathbb{E}[D_k|D_{k-1}, \ldots, D_k] = 0$) and satisfies

$$\exp\left(\frac{D_k^2}{4\gamma_k^2\|\nabla f(x^k)\|_1^2}\right) = \exp\left(\frac{(\phi_k - \psi_k)^2}{4}\right) \leq e.$$

Applying Measure Concentration Lemma 3 to MSD $D_k$ with $\sigma_k^2 = 4\gamma_k^2\|\nabla f(x^k)\|_1^2$, we derive the bound for all $\lambda > 0$ with probability at least $1 - \delta$:

$$\sum_{k=1}^T \gamma_k(\psi_k - 3\lambda\gamma_k\|\nabla f(x^k)\|_1)\|\nabla f(x^k)\|_1 \leq \Delta_1 + \frac{Ld}{2}\sum_{k=0}^{T-1}\gamma_k^2 + \frac{1}{\lambda}\log(1/\delta).$$

We use norm relation (17) and $L$-smoothness (As.2) to estimate maximum gradient norm for all $k \in \overline{2, T+1}$:

$$
\begin{aligned}
\|\nabla f(x^k)\|_1 &\leq \sqrt{d}\|\nabla f(x^k)\|_2 \leq \sqrt{d}\|\nabla f(x^k) - \nabla f(x^{k-1}) + \nabla f(x^{k-1})\|_2 \\
&\leq \sqrt{d}\|\nabla f(x^k) - \nabla f(x^{k-1})\|_2 + \sqrt{d}\|\nabla f(x^{k-1})\|_2 \leq \sqrt{d}L\|x^k - x^{k-1}\|_2 + \sqrt{d}\|\nabla f(x^{k-1})\|_2 \\
&\leq \sqrt{d}L\gamma_{k-1}\sqrt{d} + \sqrt{d}\|\nabla f(x^{k-1})\|_2 \leq \sqrt{d}\|\nabla f(x^1)\|_1 + Ld\sum_{\tau=1}^{k-1}\gamma_\tau. 
\end{aligned}
\tag{24}
$$

Hence, the choice $\lambda := \frac{1}{6d(\gamma^{max}\|\nabla f(x^1)\|_1 + C_T L)}$ where $C_T := \max_{k \in \overline{1,T}} \gamma_k \cdot \sum_{\tau=1}^{k-1}\gamma_\tau$ and $\gamma^{max} := \max_{k \in \overline{1,T}} \gamma_k$ yields with probability at least $1 - \delta$:

$$
\sum_{k=1}^{T}\gamma_k\left(\psi_k - \frac{1}{2}\right)\|\nabla f(x^k)\|_1 \leq \Delta_1 + \frac{Ld}{2}\sum_{k=1}^{T}\gamma_k^2 + 6d(\gamma^{max}\|\nabla f(x^1)\|_1 + C_T L)\log(1/\delta),
\tag{25}
$$

Next, we estimate each term $\psi_k\|\nabla f(x^k)\|_1$ in the previous sum:

$$
\begin{aligned}
\psi_k\|\nabla f(x^k)\|_1 &= \mathbb{E}\left[\langle \nabla f(x^k), \mathrm{sign}(g^k)\rangle | x^k\right] \\
&= \|\nabla f(x^k)\|_1 - \sum_{i=1}^{d} 2|\nabla f(x^k)|_i \cdot \mathbb{P}(\mathrm{sign}(\nabla f(x^k))_i \neq \mathrm{sign}(g^k)_i | x^k).
\end{aligned}
\tag{26}
$$

For each coordinate, we have a bound derived from Markov's inequality (19) followed by Jensen's inequality (18):

$$
\begin{aligned}
\mathbb{P}(\mathrm{sign}(\nabla f(x^k))_i \neq \mathrm{sign}(g^k)_i | x^k) &\leq \mathbb{P}(|\nabla f(x^k)_i - g_i^k| \geq |\nabla f(x^k)_i| | x^k) \leq \frac{\mathbb{E}_{\xi^k}[|\nabla f(x^k)_i - g_i^k|]}{|\nabla f(x^k)_i|} \\
&\leq \frac{(\mathbb{E}_{\xi^k}[|\nabla f(x^k)_i - g_i^k|^\kappa])^{\frac{1}{\kappa}}}{|\nabla f(x^k)_i|} \leq \frac{\sigma_i}{|\nabla f(x^k)_i|}.
\end{aligned}
\tag{27}
$$

Hence, the whole sum can be bounded as

$$
\sum_{i=1}^{d} 2|\nabla f(x^k)|_i \cdot \mathbb{P}(\mathrm{sign}(\nabla f(x^k))_i \neq \mathrm{sign}(g^k)_i | x^k) \leq 2\|\vec{\sigma}\|_1.
$$

Finally, we put this bound in (25) and obtain:

$$
\begin{aligned}
\frac{1}{2}\sum_{k=1}^{T}\gamma_k\|\nabla f(x^k)\|_1 &\leq f(x^1) - f(x^*) + \frac{Ld}{2}\sum_{k=1}^{T}\gamma_k^2 + 2\sum_{k=1}^{T}\gamma_k\|\vec{\sigma}\|_1 \\
&\quad + 6d(\gamma^{max}\|\nabla f(x^1)\|_1 + C_T L)\log(1/\delta).
\end{aligned}
\tag{28}
$$

Plugging in constant stepsizes $\gamma_k \equiv \gamma$ implies $C_T = T\gamma^2, \gamma^{max} = \gamma$ and the required inequality (2):

$$
\frac{1}{T}\sum_{k=1}^{T}\|\nabla f(x^k)\|_1 \leq \frac{2\Delta_1}{T\gamma} + 16Ld\gamma\log(1/\delta) + 4\|\vec{\sigma}\|_1 + 12\frac{d\|\nabla f(x^1)\|_1}{T}\log(1/\delta).
$$

$\square$

### C.3. Proof of minibatch-SignSGD Complexity Theorem 1

*Proof.* According to Lemma 1, minibatch-SignSGD with batched gradient estimates of batchsize $B$ corrupted by HT noise with $\vec{\sigma_B}$ convergence as follows:

$$
\frac{1}{T}\sum_{k=1}^{T}\|\nabla f(x^k)\|_1 \leq \frac{2\Delta_1}{T\gamma} + 16Ld\gamma\log(1/\delta) + 4\|\vec{\sigma_B}\|_1 + 12\frac{d\|\nabla f(x^1)\|_1}{T}\log(1/\delta).
$$

Due to Batching Lemma 4, we can estimate the $\kappa-$th moment of the batched estimate as $\|\vec{\sigma_B}\|_1 \leq \frac{2\|\vec{\sigma}\|_1}{B^{\frac{\kappa-1}{\kappa}}}$ and derive:

$$\frac{1}{T}\sum_{k=1}^{T}\|\nabla f(x^k)\|_1 \leq \frac{2\Delta_1}{T\gamma} + 16Ld\gamma\log(1/\delta) + 8\frac{\|\vec{\sigma}\|_1}{B^{\frac{\kappa-1}{\kappa}}} + 12\frac{d\|\nabla f(x^1)\|_1}{T}\log(1/\delta). \tag{29}$$

We can omit the last term since its dependency on $T$ has the largest power.

**1) For arbitrary tuning**, we use parameters $T, \gamma_k = \frac{\gamma_0}{\sqrt{T}}, B_k = \max\{1, B_0 T\}$ to get:

$$\frac{1}{T}\sum_{k=1}^{T}\|\nabla f(x^k)\|_1 \leq \frac{2\Delta_1}{\sqrt{T}\gamma_0} + 16\frac{Ld\gamma_0}{\sqrt{T}}\log(1/\delta) + 8\frac{\|\vec{\sigma}\|_1}{B_0^{\frac{\kappa-1}{\kappa}}T^{\frac{\kappa-1}{\kappa}}} + 12\frac{d\|\nabla f(x^1)\|_1}{T}\log(1/\delta).$$

Setting such $T$ that the first two terms become less than $\varepsilon$, we obtain the final complexity $N = T \cdot B_0 T$.

**2) For optimal tuning**, we first choose large enough $B$ to bound the term $8\frac{\|\vec{\sigma}\|_1}{B^{\frac{\kappa-1}{\kappa}}} \leq \varepsilon/2 \Rightarrow B_k \equiv \max\left\{1, \left(\frac{16\|\vec{\sigma}\|_1}{\varepsilon}\right)^{\frac{\kappa}{\kappa-1}}\right\}$.
Then we choose optimal $\gamma = \sqrt{\frac{\Delta_1}{8L_\delta dT}}$ minimizing $\min_\gamma\left\{\frac{2\Delta_1}{T\gamma} + 16Ld\gamma\log(1/\delta)\right\} = \sqrt{\frac{128\Delta_1 Ld\log(1/\delta)}{T}}$. Finally, $T$ is set to bound $\sqrt{\frac{128\Delta_1 Ld\log(1/\delta)}{T}} \leq \varepsilon/2 \Rightarrow T = O\left(\frac{\Delta_1 L\log(1/\delta)d}{\varepsilon^2}\right)$. $\qquad\square$

### C.4. Proof of M-SignSGD Complexity Theorem 3

In this proof, we generalize Theorem 1 from (Sun et al., 2023) for HT noise.

*Proof.* Since we set constant steps sizes and momentum, we denote them as $\gamma \equiv \gamma_k$ and $\beta \equiv \beta_k$, respectively. We use notations $\epsilon^k := m^k - \nabla f(x^k)$ and $\theta^k := g^k - \nabla f(x^k)$. Therefore, we have at $k$-th step values:

$$\begin{aligned}
m^k &= \beta m^{k-1} + (1-\beta)g^k = \gamma(\epsilon^{k-1} + \nabla f(x^{k-1})) + (1-\gamma)(\theta^k + \nabla f(x^k)), \\
\epsilon^k &= m^k - \nabla f(x^k) = \beta\epsilon^{k-1} + \beta(\underbrace{\nabla f(x^{k-1}) - \nabla f(x^k)}_{=:s^k}) + (1-\beta)\theta^k, \\
\epsilon^k &= m^k - \nabla f(x^k) = \beta\epsilon^{k-1} + \beta s^k + (1-\beta)\theta^k.
\end{aligned}$$

Unrolling the recursion, we obtain an explicit formula (upper index of $\beta$ is its power):

$$\epsilon^k = \beta^{k-1}\epsilon^1 + \sum_{i=2}^{k}\beta^{k-i}s^i + (1-\beta)\sum_{i=2}^{k}\beta^{k-i}\theta^i. \tag{30}$$

From smoothness of $f$ (As. 2) follows the bound:

$$\|s^k\|_2 \leq L\|x^{k-1} - x^k\|_2 \leq L\sqrt{d}\gamma.$$

Hence, the norm of (30) can be bounded as:

$$\|\epsilon^k\|_2 \leq \beta^{k-1}\|\epsilon^1\|_2 + L\sqrt{d}\gamma \cdot \sum_{i=2}^{k}\beta^{k-i} + (1-\beta)\|\sum_{i=2}^{k}\beta^{k-i}\theta^i\|_2.$$

We notice that variables $\{\theta_i\}$ are martingale difference sequence from Lemma 4 which we plan to use. Due to the formal definition of $\theta^i = g^i - \nabla f(x^i) = \nabla f(x^i, \xi_i) - \nabla f(x^i)$ and M-SinSGD step, the conditioning on $\theta^{i-1}, \ldots, \theta^1$ with randomness $\xi_1, \ldots, \xi_{i-1}$ is equivalent to the conditioning on point s $x^i, \ldots, x^2$. Hence, we show by definition of martingale difference sequence that

$$\mathbb{E}[\theta^i|\theta^{i-1}, \ldots, \theta^1] = \mathbb{E}[\theta^i|x^i, \ldots, x^2] = \mathbb{E}[\nabla f(x^i, \xi_i) - \nabla f(x^i)|x^i, \ldots, x^2] = 0.$$

To take math expectation from both sides, we first take it from the term

$$\mathbb{E}\left[\|\sum_{i=2}^{k}\beta^{k-i}\theta^i\|_2\right] \le \left(\mathbb{E}\left[\|\sum_{i=2}^{k}\beta^{k-i}\theta^i\|_2^\kappa\right]\right)^{\frac{1}{\kappa}} \overset{\text{Lem. }4}{\le} \left(\sum_{i=2}^{k}2\mathbb{E}\left[\|\beta^{(k-i)}\theta^i\|_2^\kappa\right]\right)^{\frac{1}{\kappa}} \le \left(\sum_{i=2}^{k}2\beta^{\kappa(k-i)}\mathbb{E}\left[\|\theta^i\|_2^\kappa\right]\right)^{\frac{1}{\kappa}}. \quad (31)$$

For each $i \in \overline{2,T}$, we estimate $\mathbb{E}\left[\|\theta^i\|_2^\kappa\right]$ as

$$\mathbb{E}\left[\|\theta^i\|_2^\kappa\right] \overset{(17)}{\le} \mathbb{E}\left[\|\theta^i\|_\kappa^\kappa\right] = \mathbb{E}\left[\sum_{j=1}^{d}|g_j^k - \nabla f(x^k)_j|^\kappa\right] \overset{As.3}{\le} \sum_{j=1}^{d}\sigma_j^\kappa = \|\vec{\sigma}\|_\kappa^\kappa. \quad (32)$$

We continue bounding (31) with

$$(31) \le \left(\sum_{i=2}^{k}2\beta^{\kappa(k-i)}\|\vec{\sigma}\|_\kappa^\kappa\right)^{\frac{1}{\kappa}} \le \frac{2\|\vec{\sigma}\|_\kappa}{(1-\beta^\kappa)^{\frac{1}{\kappa}}}.$$

Therefore, the final math expectation can be calculated as:

$$\mathbb{E}\|\epsilon^k\|_2 \quad \le \quad \beta^{k-1}\mathbb{E}\|\epsilon^1\|_2 + \frac{L\sqrt{d}\gamma}{1-\beta} + \frac{2(1-\beta)\|\vec{\sigma}\|_\kappa}{(1-\beta^\kappa)^{\frac{1}{\kappa}}}. \quad (33)$$

Now, we can use update step Lemma 5 and then take math expectation:

$$f(x^{k+1}) - f(x^k) \quad \le \quad -\gamma\|\nabla f(x^k)\|_1 + 2\gamma\sqrt{d}\|\epsilon^k\|_2 + \frac{L\gamma^2 d}{2},$$

$$\mathbb{E}[f(x^{k+1})] - \mathbb{E}[f(x^k)] \quad \le \quad -\gamma\mathbb{E}[\|\nabla f(x^k)\|_1] + 2\gamma\sqrt{d}\beta^{k-1}\mathbb{E}\|\epsilon^1\|_2 + \frac{2Ld\gamma^2}{1-\beta} + \frac{4\gamma\sqrt{d}(1-\beta)\|\vec{\sigma}\|_\kappa}{(1-\beta^\kappa)^{\frac{1}{\kappa}}} + \frac{L\gamma^2 d}{2}.$$

Summing it over $k$ and dividing by $T\gamma$, we derive

$$\frac{1}{T}\sum_{k=1}^{T}\mathbb{E}\|\nabla f(x^k)\|_1 \le \frac{f(x^1) - f_*}{\gamma T} + \frac{4Ld\gamma}{1-\beta} + \frac{4\sqrt{d}(1-\beta)\|\vec{\sigma}\|_\kappa}{(1-\beta^\kappa)^{\frac{1}{\kappa}}} + 2\sqrt{d}\sum_{k=1}^{T}\beta^{k-1}\|\epsilon^0\|_2/T. \quad (34)$$

We omit the last term since its dependency on $T$ is much weaker.

**1) For arbitrary tuning**, we set $1-\beta = \frac{1}{\sqrt{T}}, \gamma = \gamma_0 T^{-\frac{3}{4}}$ and obtain

$$\frac{1}{T}\sum_{k=1}^{T}\mathbb{E}\|\nabla f(x^k)\|_1 \le \frac{\Delta_1}{\gamma_0 T^{\frac{1}{4}}} + \frac{4Ld\gamma_0}{T^{\frac{1}{4}}} + \frac{4\sqrt{d}\|\vec{\sigma}\|_\kappa}{T^{\frac{\kappa-1}{2\kappa}}} + \frac{2\sqrt{d}\|\epsilon^0\|_2}{T^{\frac{1}{2}}}.$$

Next, we choose $T$ to limit $\frac{\Delta_1/\gamma_0 + Ld\gamma_0}{T^{\frac{1}{4}}} \le \frac{\varepsilon}{2}$ and $\frac{4\sqrt{d}\|\vec{\sigma}\|_1}{T^{\frac{\kappa-1}{2\kappa}}} \le \frac{\varepsilon}{2}$.

**2) For optimal tuning**, we first choose optimal $\gamma = \sqrt{\frac{\Delta_1(1-\beta)}{4LdT}}$ via minimizing $\min_\gamma\left\{\frac{\Delta_1}{\gamma T} + \frac{4Ld\gamma}{1-\beta}\right\}$. Then we find optimal $\beta = 1 - \min\left\{1, \left(\frac{\Delta_1 L}{\|\vec{\sigma}\|_\kappa^2 T}\right)^{\frac{\kappa}{3\kappa-2}}\right\}$ via minimizing remaining

$$\min_\beta\left\{\sqrt{\frac{16\Delta_0 Ld}{T(1-\beta)}} + \frac{4\sqrt{d}(1-\beta)\|\vec{\sigma}\|_\kappa}{(1-\beta^\kappa)^{\frac{1}{\kappa}}}\right\}.$$

Finally, we select $T$ according to required accuracy $\varepsilon$. $\qquad\qquad\square$

## C.5. Proofs of MajorityVote-SignSGD Complexity Theorems 2 and 5

*Proof of Theorem 2.* The beginning of this proof exactly copies the proof of SignSGD Convergence Lemma (Appendix C.2) until equality (26). We have to estimate the probability of failure of majority voting for each coordinate $j$ conditioned on $x^k$, namely,

$$\mathbb{P}\left(\text{sign}(\nabla f(x^k))_j \neq \text{sign}\left[\sum_{i=1}^{M} \text{sign}(g_i^k)\right]_j\right), \quad g_i^k = \nabla f(x^k, \xi_i^k).$$

We use the generalized Gauss's Inequality about distribution of unimodal symmetric random variables (Dharmadhikari & Joag-Dev, 1986, Theorem 1).

**Lemma 6** (Gauss's Inequality). *Let a random variable $\xi$ be unimodal symmetric with mode $\nu$ and bounded $\kappa$-th moment, $\kappa \in (1, 2]$. Then the following bound holds:*

$$\mathbb{P}[|\xi - \nu| \geq \tau] \leq \begin{cases} \left(\frac{\kappa}{\kappa+1}\right)^\kappa \frac{\mathbb{E}[|\xi-\nu|]^\kappa}{\tau^\kappa}, & \tau^\kappa \geq \frac{\kappa^\kappa}{(\kappa+1)^{\kappa-1}} \cdot \mathbb{E}[|\xi - \nu|^\kappa], \\ 1 - \left[\frac{\tau^\kappa}{(\kappa+1)\mathbb{E}[|\xi-\nu|]^\kappa}\right]^{\frac{1}{\kappa}}, & \tau^\kappa \leq \frac{\kappa^\kappa}{(\kappa+1)^{\kappa-1}} \cdot \mathbb{E}[|\xi - \nu|^\kappa]. \end{cases}$$

We use Gauss's Inequality for each variable $g_{i,j}^k = \nabla f(x^k, \xi_i^k)_j$ satisfying the symmetry requirement from the theorem's statement. We denote $S_j := \frac{|\nabla f(x^k)_j|}{\sigma_j}$ and bound

$$
\begin{aligned}
\mathbb{P}\left[\text{sign}(\nabla f(x^k)_j) \neq \text{sign}(g_{i,j}^k)\right] &= \mathbb{P}\left[g_{i,j}^k - \nabla f(x^k)_j \geq |\nabla f(x^k)_j|\right] \\
&= \frac{1}{2}\mathbb{P}\left[|g_{i,j}^k - \nabla f(x^k)_j| \geq |\nabla f(x^k)_j|\right] \\
&\leq \begin{cases} \frac{1}{2}\left(\frac{\kappa}{\kappa+1}\right)^\kappa \frac{\sigma_j^\kappa}{|\nabla f(x^k)_j|^\kappa}, & |\nabla f(x^k)_j|^\kappa \geq \frac{\kappa^\kappa}{(\kappa+1)^{\kappa-1}} \cdot \sigma_j^\kappa, \\ \frac{1}{2} - \frac{1}{2}\left[\frac{|\nabla f(x^k)_j|^\kappa}{(\kappa+1)\sigma_j^\kappa}\right]^{\frac{1}{\kappa}}, & |\nabla f(x^k)_j|^\kappa \leq \frac{\kappa^\kappa}{(\kappa+1)^{\kappa-1}} \cdot \sigma_j^\kappa, \end{cases} \\
&\leq \begin{cases} \frac{1}{2}\left(\frac{\kappa}{\kappa+1}\right)^\kappa \frac{1}{S_j^\kappa}, & S_j^\kappa \geq \frac{\kappa^\kappa}{(\kappa+1)^{\kappa-1}}, \\ \frac{1}{2} - \frac{1}{2}\frac{S_j}{(\kappa+1)^{\frac{1}{\kappa}}}, & S_j^\kappa \leq \frac{\kappa^\kappa}{(\kappa+1)^{\kappa-1}}, \end{cases}
\end{aligned}
$$

We denote probability of failure of a single estimate by

$$
\begin{aligned}
q_j &:= \mathbb{P}\left[\text{sign}(\nabla f(x^k)_j) \neq \text{sign}(g_{i,j}^k)\right] \\
&\leq \begin{cases} \frac{1}{2}\left(\frac{\kappa}{\kappa+1}\right)^\kappa \frac{1}{S_j^\kappa}, & S_j^\kappa \geq \frac{\kappa^\kappa}{(\kappa+1)^{\kappa-1}}, \\ \frac{1}{2} - \frac{1}{2}\frac{S_j}{(\kappa+1)^{\frac{1}{\kappa}}}, & S_j^\kappa \leq \frac{\kappa^\kappa}{(\kappa+1)^{\kappa-1}}, \end{cases} \\
&=: \tilde{q}_j(S_j). \tag{35}
\end{aligned}
$$

Moreover, this probability $q_j \leq \tilde{q}_j(S_j) < \frac{1}{2}$, and the deviation of $q_j$ from $\frac{1}{2}$ can be bounded by

$$\varepsilon_j := \frac{1}{2} - q_j \leq \frac{1}{2} - \tilde{q}_j(S_j) =: \tilde{\varepsilon}_j(S_j).$$

The probability of getting the wrong sign can be restated as the probability of failing half out of $M$ Bernoulli trials with fail probability $q_j$:

$$\mathbb{P}\left[\text{sign}(\nabla f(x^k)_j) \neq \text{sign}\left[\sum_{i=1}^{M} \text{sign}(g_{i,j}^k)\right]\right] \leq \frac{1}{1 + \frac{M}{\frac{1}{4\varepsilon_j^2}-1}} < \frac{1}{1 + \frac{M}{\frac{1}{4\tilde{\varepsilon}_j^2(S_j)}-1}}. \tag{36}$$

First, we consider the case $S_j \geq \frac{\kappa}{(\kappa+1)^{\frac{\kappa-1}{\kappa}}}$:

$$\frac{1}{4\tilde{\varepsilon}_j^2(S_j)} - 1 = \frac{1}{4\left(\frac{1}{2} - \frac{1}{2}\left(\frac{\kappa}{\kappa+1}\right)^\kappa \frac{1}{S_j^\kappa}\right)^2} - 1 = \frac{1}{1 + \left(\frac{\kappa}{\kappa+1}\right)^{2\kappa} \frac{1}{S_j^{2\kappa}} - 2\left(\frac{\kappa}{\kappa+1}\right)^\kappa \frac{1}{S_j^\kappa}} - 1$$

$$= \frac{1}{S_j^\kappa} \frac{2\left(\frac{\kappa}{\kappa+1}\right)^\kappa - \left(\frac{\kappa}{\kappa+1}\right)^\kappa \frac{1}{S_j^\kappa}}{1 + \left(\frac{\kappa}{\kappa+1}\right)^\kappa \frac{1}{S_j^\kappa} - 2\left(\frac{\kappa}{\kappa+1}\right)^\kappa \frac{1}{S_j^\kappa}}$$

$$\leq \frac{1}{S_j^\kappa} \frac{2\left(\frac{\kappa}{\kappa+1}\right)^\kappa}{1 - 2\left(\frac{\kappa}{\kappa+1}\right)^\kappa \frac{1}{S_j^\kappa}} \leq \frac{1}{S_j^\kappa} \frac{\kappa+1}{\kappa-1}. \tag{37}$$

We use the inequality $\frac{1}{1+x^\kappa} \leq \frac{1}{x}, x > 0$ on (36):

$$(36) \leq \frac{\left(\frac{1}{4\tilde{\varepsilon}_j^2(S_j)} - 1\right)^{\frac{1}{\kappa}}}{M^{\frac{1}{\kappa}}} \leq \left(\frac{\kappa+1}{\kappa-1}\right)^{\frac{1}{\kappa}} \cdot \frac{1}{M^{\frac{1}{p}}} \cdot \frac{1}{S_j}. \tag{38}$$

For the case $S_j < \frac{\kappa}{(\kappa+1)^{\frac{\kappa-1}{\kappa}}}$, we derive the bound:

$$\frac{1}{4\tilde{\varepsilon}_j^2(S_j)} - 1 = \frac{(\kappa+1)^{\frac{2}{\kappa}}}{S_j^2} - 1 \leq \frac{4}{S_j^2}. \tag{39}$$

And we use the inequality $\frac{1}{1+x^2} \leq \frac{1}{2x}, x > 0$ on (36):

$$(36) \leq \frac{\sqrt{\frac{1}{4\tilde{\varepsilon}_j^2(S_j)} - 1}}{2\sqrt{M}} \leq \frac{1}{\sqrt{M}} \cdot \frac{1}{S_j}. \tag{40}$$

Combining (38) and (40) together, we obtain the bound for each coordinate:

$$\mathbb{P}\left[\text{sign}(\nabla f(x^k)_j) \neq \text{sign}\left[\sum_{i=1}^M \text{sign}(g_{i,j}^k)\right]\right] \leq \left(\frac{\kappa+1}{\kappa-1}\right)^{\frac{1}{\kappa}} \cdot \frac{1}{\sqrt{M}} \cdot \frac{1}{S_j} = \left(\frac{\kappa+1}{\kappa-1}\right)^{\frac{1}{\kappa}} \cdot \frac{1}{\sqrt{M}} \frac{\sigma_j}{|\nabla f(x^k)_j|}. \tag{41}$$

The rest of this proof is copying the proof of SignSGD Convergence Lemma (Appendix C.2) until the equality (26). There we replace probability of single estimate with the majority voting and obtain:

$$\sum_{j=1}^d |\nabla f(x^k)|_j \cdot \mathbb{P}\left[\text{sign}(\nabla f(x^k)_j) \neq \text{sign}\left[\sum_{i=1}^M \text{sign}(g_{i,j}^k)\right]\right] \leq \left(\frac{\kappa+1}{\kappa-1}\right)^{\frac{1}{\kappa}} \cdot \frac{\|\vec{\sigma}\|_1}{\sqrt{M}}$$

instead of

$$\sum_{j=1}^d |\nabla f(x^k)|_j \cdot \mathbb{P}(\text{sign}([\nabla f(x^k)])_j \neq [\text{sign}(g^k)]_j) \leq \|\vec{\sigma}\|_1.$$

Hence, the final bound on sum of $\ell_1$-norm of gradients with probability at least $1 - \delta$ is

$$\frac{1}{2}\sum_{k=1}^T \gamma_k \|\nabla f(x^k)\|_1 \leq f(x^1) - f(x^*) + \frac{Ld}{2}\sum_{k=1}^T \gamma_k^2 + 2\left(\frac{\kappa+1}{\kappa-1}\right)^{\frac{1}{\kappa}}\sum_{k=1}^T \gamma_k \cdot \frac{\|\vec{\sigma}\|_1}{\sqrt{M}}$$

$$+ 6d(\gamma^{max}\|\nabla f(x^1)\|_1 + C_T L)\log(1/\delta).$$

Plugging in constant stepsizes $\gamma_k \equiv \gamma$ implies $C_T = T\gamma^2, \gamma^{max} = \gamma$ and denoting $a_\kappa := \left(\frac{\kappa+1}{\kappa-1}\right)^{\frac{1}{\kappa}}$, we have :

$$\frac{1}{T}\sum_{k=1}^T \|\nabla f(x^k)\|_1 \leq \frac{2\Delta_1}{T\gamma} + 16Ld\gamma\log(1/\delta) + 4a_\kappa\|\vec{\sigma}\|_1/\sqrt{M} + 12\frac{d\|\nabla f(x^1)\|_1}{T}\log(1/\delta). \tag{42}$$

**1) For arbitrary tuning**, we use parameters $T, \gamma_k = \frac{\gamma_0}{\sqrt{T}}, M_k = \max\{1, M_0 T\}$ to get:

$$\frac{1}{T}\sum_{k=1}^{T}\|\nabla f(x^k)\|_1 \leq \frac{2\Delta_1}{\sqrt{T}\gamma_0} + 16\frac{Ld\gamma_0}{\sqrt{T}}\log(1/\delta) + 4a_\kappa\frac{\|\vec{\sigma}\|_1}{\sqrt{M_0 T}} + 12\frac{d\|\nabla f(x^1)\|_1}{T}\log(1/\delta).$$

Setting such $T$ that the first three terms become less than $\varepsilon$, we obtain the final complexity $N = T \cdot M_0 T$.

**2) For optimal tuning**, we first choose large enough $M$ to bound the term $4a_\kappa\frac{\|\vec{\sigma}\|_1}{\sqrt{M_k}} \leq \varepsilon/2 \Rightarrow M_k \equiv \max\left\{1, \left(\frac{8a_k\|\vec{\sigma}\|_1}{\varepsilon}\right)^2\right\}$. Then we choose optimal $\gamma = \sqrt{\frac{\Delta_1}{8L_\delta dT}}$ minimizing $\min_\gamma\left\{\frac{2\Delta_1}{T\gamma} + 16Ld\gamma\log(1/\delta)\right\} = \sqrt{\frac{128\Delta_1 Ld\log(1/\delta)}{T}}$. Finally, $T$ is set to bound $\sqrt{\frac{128\Delta_1 Ld\log(1/\delta)}{T}} \leq \varepsilon/2 \Rightarrow T = O\left(\frac{\Delta_1 L\log(1/\delta)d}{\varepsilon^2}\right)$. $\qquad\square$

*Proof of Theorem 5.* This proof completely copies Proof of minibatch-SignSGD Complexity Theorem starting with line (42) and substituting $\|\vec{\sigma}\|_1$ with $\frac{a_\kappa\|\vec{\sigma}\|_1}{\sqrt{M}}$. $\qquad\square$

## C.6. Proof of CompSGD Convergence Lemma 2

*Proof.* Consider the $k$-th step of CompSGD. We use smoothness of function $f$ (Lemma 1) to estimate:

$$
\begin{aligned}
f(x^{k+1}) - f(x^k) &\leq \langle\nabla f(x^k), x^{k+1} - x^k\rangle + \frac{L}{2}\|x^{k+1} - x^k\|_2^2 \\
&= -\gamma_k \cdot \text{sign}(f(x^k + \gamma_k\mathbf{e}^k, \xi_+^k) - f(x^k - \gamma_k\mathbf{e}^k, \xi_-^k)) \cdot \langle\nabla f(x^k), \mathbf{e}^k\rangle + \frac{L}{2}\gamma_k^2\|\mathbf{e}^k\|_2^2 \\
&\overset{As.5}{\leq} -\gamma_k\frac{\text{sign}(f(x^k + \gamma_k\mathbf{e}^k, \xi_+^k) - f(x^k - \gamma_k\mathbf{e}^k, \xi_-^k)) \cdot \langle\nabla f(x^k), \mathbf{e}^k\rangle}{\|\nabla f(x^k)\|_p} \cdot \|\nabla f(x^k)\|_p + \frac{L}{2}\gamma_k^2.
\end{aligned}
$$

Consequently, after summing $T$ steps, we obtain

$$\sum_{k=1}^{T}\gamma_k\frac{\text{sign}(f(x^k + \gamma_k\mathbf{e}^k, \xi_+^k) - f(x^k - \gamma_k\mathbf{e}^k, \xi_-^k)) \cdot \langle\nabla f(x^k), \mathbf{e}^k\rangle}{\|\nabla f(x^k)\|_p} \cdot \|\nabla f(x^k)\|_p \leq \underbrace{f(x^1) - f(x^*)}_{=\Delta_1} + \frac{L}{2}\sum_{k=1}^{T}\gamma_k^2. \quad (43)$$

Next, we deal with terms $\phi_k := \frac{\text{sign}(f(x^k+\gamma_k\mathbf{e}^k,\xi_+^k)-f(x^k-\gamma_k\mathbf{e}^k,\xi_-^k))\cdot\langle\nabla f(x^k),\mathbf{e}^k\rangle}{\|\nabla f(x^k)\|_p}$, $\psi_k := \mathbb{E}[\phi_k|x^k]$ and $D_k := -\gamma_k(\phi_k - \psi_k)\|\nabla f(x^k)\|_p/\alpha_p$, where $\alpha_p$ is taken from lemma's statement. The terms $\phi_k$ are bounded with $|\phi_k| \leq 1$ due to Cauchy–Schwarz inequality :

$$|\phi_k| = \frac{|\text{sign}(f(x^k + \gamma_k\mathbf{e}^k, \xi_+^k) - f(x^k - \gamma_k\mathbf{e}^k, \xi_-^k)) \cdot \langle\nabla f(x^k), \mathbf{e}^k\rangle|}{\|\nabla f(x^k)\|_p} \leq \frac{|\langle\nabla f(x^k), \mathbf{e}^k\rangle|}{\|\nabla f(x^k)\|_p} \leq \|\mathbf{e}^k\|_q \overset{As.5}{\leq} 1.$$

We note that $D_k$ is a martingale difference sequence ($\mathbb{E}[D_k|D_{k-1}, \ldots, D_1] = 0$) satisfying the inequality

$$\exp\left(\frac{D_k^2}{4\gamma_k^2\|\nabla f(x^k)\|_p^2/\alpha_p^2}\right) = \exp\left(\frac{(\phi_k - \psi_k)^2}{4}\right) \leq e.$$

Applying Measure Concentration Lemma 3 to MSD $D_k$ with $\sigma_k^2 = 4\gamma_k^2\|\nabla f(x^k)\|_p^2/\alpha_p$, we derive the bound for all $\lambda > 0$ with probability at least $1 - \delta$ :

$$\sum_{k=1}^{T}\gamma_k(\psi_k - 3\lambda\gamma_k\|\nabla f(x^k)\|_p)\frac{\|\nabla f(x^k)\|_p}{\alpha_p} \leq \frac{\Delta_1}{\alpha_p} + \frac{L}{2\alpha_p}\sum_{k=1}^{T}\gamma_k^2 + \frac{1}{\lambda}\log(1/\delta)$$

Next, we use norm relation (17), $L$-smoothness (As. 2) and update step of CompSGD to estimate maximal norm achieved for $k \in \overline{2, T+1}$:

$$
\begin{aligned}
\|\nabla f(x^k)\|_p &\leq d^{\frac{1}{p}-\frac{1}{2}}\|\nabla f(x^k)\|_2 \leq d^{\frac{1}{p}-\frac{1}{2}}\|\nabla f(x^k) - \nabla f(x^{k-1}) + \nabla f(x^{k-1})\|_2 \\
&\leq d^{\frac{1}{p}-\frac{1}{2}}\|\nabla f(x^k) - \nabla f(x^{k-1})\|_2 + d^{\frac{1}{p}-\frac{1}{2}}\|\nabla f(x^{k-1})\|_2 \leq d^{\frac{1}{p}-\frac{1}{2}}L\|x^k - x^{k-1}\|_2 + d^{\frac{1}{p}-\frac{1}{2}}\|\nabla f(x^{k-1})\|_2 \\
&\leq d^{\frac{1}{p}-\frac{1}{2}}L\gamma_{k-1} + d^{\frac{1}{p}-\frac{1}{2}}\|\nabla f(x^{k-1})\|_2 \leq d^{\frac{1}{p}-\frac{1}{2}}\|\nabla f(x^1)\|_2 + d^{\frac{1}{p}-\frac{1}{2}}L\sum_{\tau=1}^{k-1}\gamma_\tau. \quad (44)
\end{aligned}
$$

Hence, the choice $\lambda := \frac{\alpha_p}{6d^{\frac{1}{p}-\frac{1}{2}}(\gamma^{max}\|\nabla f(x^1)\|_2 + C_T L)}$ yields with probability at least $1 - \delta$:

$$\sum_{k=1}^{T} \gamma_k \left( \frac{\psi_k}{\alpha_p} - \frac{1}{2} \right) \|\nabla f(x^k)\|_p \leq \frac{\Delta_1}{\alpha_p} + \frac{L}{2\alpha_p} \sum_{k=1}^{T} \gamma_k^2 + \frac{6d^{\frac{1}{p}-\frac{1}{2}}}{\alpha_p}(\gamma^{max}\|\nabla f(x^1)\|_2 + C_T L)\log(1/\delta), \tag{45}$$

where $C_T := \max_{k \in \overline{1,T}} \gamma_k \cdot \sum_{\tau=1}^{k-1} \gamma_\tau$. Finally, we estimate the term $\psi_k\|\nabla f(x^k)\|_p$:

$$\mathbb{E}_{\xi,\mathbf{e}^k} \left[ \text{sign}(f(x^k + \gamma_k\mathbf{e}^k, \xi_+^k) - f(x^k - \gamma_k\mathbf{e}^k, \xi_-^k)) \cdot \langle \nabla f(x^k), \mathbf{e}^k \rangle \right] = \mathbb{E}_{\mathbf{e}^k}|\langle \nabla f(x^k), \mathbf{e}^k \rangle|$$
$$- \quad \mathbb{E}_{\mathbf{e}^k} \left[ 2 \cdot \mathbb{P}_\xi \left[ \text{sign}(f(x^k + \gamma_k\mathbf{e}^k, \xi_+^k) - f(x^k - \gamma_k\mathbf{e}^k, \xi_-^k)) \neq \text{sign}(\langle \nabla f(x^k), \mathbf{e}^k \rangle) \right] \cdot |\langle \nabla f(x^k), \mathbf{e}^k \rangle| \right].$$

Next, we consider two cases to deal with probability over $\xi$: $|\langle \nabla f(x^k), \mathbf{e}^k \rangle| \geq 2\gamma_k L$ and $|\langle \nabla f(x^k), \mathbf{e}^k \rangle| \leq 2\gamma_k L$.

**Case** $|\langle \nabla f(x^k), \mathbf{e}^k \rangle| \leq 2\gamma_k L$:

$$\mathbb{E}_{\xi,\mathbf{e}^k} \left[ \text{sign}(f(x^k + \gamma_k\mathbf{e}^k, \xi_+^k) - f(x^k - \gamma_k\mathbf{e}^k, \xi_-^k)) \cdot \langle \nabla f(x^k), \mathbf{e}^k \rangle \right] \geq -\mathbb{E}_{\mathbf{e}^k}[|\langle \nabla f(x^k), \mathbf{e}^k \rangle|]$$
$$\geq \mathbb{E}_{\mathbf{e}^k}[|\langle \nabla f(x^k), \mathbf{e}^k \rangle|] - 4\gamma_k L$$
$$\overset{\text{As. } 5}{\geq} \alpha_p\|\nabla f(x^k)\|_p - 4\gamma_k L.$$

**Case** $|\langle \nabla f(x^k), \mathbf{e}^k \rangle| \geq 2\gamma_k L$:

We change sign operators to equivalent ones denoting $\theta_+^k := f(x^k + \gamma_k\mathbf{e}^k, \xi_+^k) - f(x^k + \gamma_k\mathbf{e}^k)$ and $\theta_-^k := f(x^k - \gamma_k\mathbf{e}^k, \xi_-^k) - f(x^k - \gamma_k\mathbf{e}^k)$:

$$\text{sign}(f(x^k + \gamma_k\mathbf{e}^k, \xi_+^k) - f(x^k - \gamma_k\mathbf{e}^k, \xi_-^k)) \neq \text{sign}(\langle \nabla f(x^k), \mathbf{e}^k \rangle)$$
$$\Updownarrow$$
$$\text{sign}(f(x^k + \gamma_k\mathbf{e}^k) - f(x^k - \gamma_k\mathbf{e}^k) + \theta_+^k - \theta_-^k) \neq \text{sign}(2\gamma_k \cdot \langle \nabla f(x^k), \mathbf{e}^k \rangle).$$

Further, we can bound probability by considering bigger number of cases:

$$\mathbb{P}_\xi \left[ \text{sign}(f(x^k + \gamma_k\mathbf{e}^k, \xi_+^k) - f(x^k - \gamma_k\mathbf{e}^k, \xi_-^k)) \neq \text{sign}(\langle \nabla f(x^k), \mathbf{e}^k \rangle) \right] \tag{46}$$
$$= \mathbb{P}_\xi \left[ \text{sign}(f(x^k + \gamma_k\mathbf{e}^k) - f(x^k - \gamma_k\mathbf{e}^k) + \theta_+^k - \theta_-^k) \neq \text{sign}(2\gamma_k \cdot \langle \nabla f(x^k), \mathbf{e}^k \rangle) \right]$$
$$\leq \mathbb{P}_\xi \left[ |f(x^k + \gamma_k\mathbf{e}^k) - f(x^k - \gamma_k\mathbf{e}^k) + \theta_+^k - \theta_-^k - 2\gamma_k \cdot \langle \nabla f(x^k), \mathbf{e}^k \rangle| \geq 2\gamma_k \cdot |\langle \nabla f(x^k), \mathbf{e}^k \rangle| \right]$$
$$\leq \mathbb{P}_\xi \left[ |f(x^k + \gamma_k\mathbf{e}^k) - f(x^k - \gamma_k\mathbf{e}^k) - 2\gamma_k \cdot \langle \nabla f(x^k), \mathbf{e}^k \rangle| + |\theta_+^k - \theta_-^k| \geq 2\gamma_k \cdot |\langle \nabla f(x^k), \mathbf{e}^k \rangle| \right]$$
$$\leq \mathbb{P}_\xi \left[ 2L^2\gamma_k^2 + |\theta_+^k - \theta_-^k| \geq 2\gamma_k \cdot |\langle \nabla f(x^k), \mathbf{e}^k \rangle| \right]. \tag{47}$$

Since we consider the case $|\langle \nabla f(x^k), \mathbf{e}^k \rangle| \geq 2\gamma_k L$, then we bound

$$(47) \quad \leq \quad \mathbb{P}_\xi \left[ \gamma_k \cdot |\langle \nabla f(x^k), \mathbf{e}^k \rangle| + |\theta_+^k - \theta_-^k| \geq 2\gamma_k \cdot |\langle \nabla f(x^k), \mathbf{e}^k \rangle| \right]$$
$$\leq \quad \mathbb{P}_\xi \left[ |\theta_+^k - \theta_-^k| \geq \gamma_k \cdot |\langle \nabla f(x^k), \mathbf{e}^k \rangle| \right]$$
$$\overset{\text{Markov ineq.}(19):}{\leq} \quad \frac{\mathbb{E}_\xi[|\theta_+^k - \theta_-^k|]}{\gamma_k \cdot |\langle \nabla f(x^k), \mathbf{e}^k \rangle|} \leq \frac{2\sigma}{\gamma_k \cdot |\langle \nabla f(x^k), \mathbf{e}^k \rangle|}. \tag{48}$$

Finally, we have can obtain the bound

$$\mathbb{E}_{\xi,\mathbf{e}^k} \left[ \text{sign}(f(x^k + \gamma_k\mathbf{e}^k, \xi_+^k) - f(x^k - \gamma_k\mathbf{e}^k, \xi_-^k)) \cdot \langle \nabla f(x^k), \mathbf{e}^k \rangle \right] \geq \mathbb{E}_{\mathbf{e}^k}|\langle \nabla f(x^k), \mathbf{e}^k \rangle| - \frac{4\sigma}{\gamma_k}$$
$$\overset{\text{As. } 5}{\geq} \alpha_p\|\nabla f(x^k)\|_p - \frac{4\sigma}{\gamma_k}.$$

Combining two cases together, we get that $\psi_k\|\nabla f(x^k)\|_p \geq \alpha_p\|\nabla f(x^k)\|_p - 4\gamma_k L - \frac{4\sigma}{\gamma_k}$, and the bound follows from (45)

$$\frac{1}{2}\sum_{k=1}^{T}\gamma_k\|\nabla f(x^k)\|_p \leq \frac{\Delta_1}{\alpha_p} + \frac{L}{2\alpha_p}\sum_{k=1}^{T}\gamma_k^2 + \sum_{k=1}^{T}\gamma_k\cdot\frac{4L\gamma_k}{\alpha_p} + 4\sum_{k=1}^{T}\frac{\sigma}{\alpha_p}$$
$$+ \frac{6d^{\frac{1}{p}-\frac{1}{2}}}{\alpha_p}(\gamma^{max}\|\nabla f(x^0)\|_2 + C_T L)\log(1/\delta)$$

Plugging in constant stepsizes $\gamma_k \equiv \gamma$, $C_T = T\gamma^2$, $\gamma^{max} = \gamma$ and dividing both sides by $\frac{T\gamma}{2}$ yields the required result:

$$\frac{1}{T}\sum_{k=1}^{T}\|\nabla f(x^k)\|_p \leq \frac{2\Delta_1}{T\alpha_p\gamma} + 24d^{\frac{1}{p}-\frac{1}{2}}\frac{L\gamma}{\alpha_p}\log(1/\delta) + \frac{8\sigma}{\alpha_p\gamma} + \frac{12d^{\frac{1}{p}-\frac{1}{2}}\|\nabla f(x^1)\|_2}{T\alpha_p}\log(1/\delta). \quad (49)$$

$\square$

## C.7. Proof of minibatch-CompSGD Complexity Theorem 6

*Proof.* We start with CompSGD Convergence Lemma 2 and constant batchsizes $B$, stepsizes $\gamma$ (49):

$$\frac{1}{T}\sum_{k=1}^{T}\|\nabla f(x^k)\|_p \leq \frac{2\Delta_1}{T\alpha_p\gamma} + 24d^{\frac{1}{p}-\frac{1}{2}}\frac{L\gamma}{\alpha_p}\log(1/\delta) + \frac{8\sigma_B}{\alpha_p\gamma} + \frac{12d^{\frac{1}{p}-\frac{1}{2}}\|\nabla f(x^1)\|_2}{T\alpha_p}\log(1/\delta).$$

Due to Batching Lemma 4, we can estimate the $\kappa-$th moment of the batched function as:

$$\sigma_B \leq \frac{2\sigma}{B^{\frac{\kappa-1}{\kappa}}}.$$

Hence, we have

$$\frac{1}{T}\sum_{k=1}^{T}\|\nabla f(x^k)\|_p \leq \frac{2\Delta_1}{T\alpha_p\gamma} + 24d^{\frac{1}{p}-\frac{1}{2}}\frac{L\gamma}{\alpha_p}\log(1/\delta) + \frac{16\sigma}{B^{\frac{\kappa-1}{\kappa}}\alpha_p\gamma} + \frac{12d^{\frac{1}{p}-\frac{1}{2}}\|\nabla f(x^1)\|_2}{T\alpha_p}\log(1/\delta).$$

**Optimal tuning:** $T$ dependency in the first three terms is dominating in comparison with the last term, hence, we neglect it. Choosing $B$ such that $\frac{\sigma}{\Delta_1 B^{\frac{\kappa-1}{\kappa}}} \leq \frac{1}{T}$, we have

$$\frac{1}{T}\sum_{k=1}^{T}\|\nabla f(x^k)\|_p \leq \frac{18\Delta_1}{T\alpha_p\gamma} + 24d^{\frac{1}{p}-\frac{1}{2}}\frac{L\gamma}{\alpha_p}\log(1/\delta).$$

With $\gamma = \sqrt{\frac{\Delta_1}{TLd^{\frac{1}{p}-\frac{1}{2}}}}$, we obtain the required number of iterations $T$ to achieve $\frac{1}{T}\sum_{k=1}^{T}\|\nabla f(x^k)\|_p \leq \varepsilon$ equals to $\tilde{O}\left(\frac{\Delta_1 L}{\varepsilon^2\alpha_p}\right)$.

$\square$

## C.8. Proof of MajorityVote-CompSGD Complexity Theorem 4

*Proof.* The beginning of the proof copies the proof of CompSGD Convergence Lemma C.6 until the line (46) where we instead estimate probability

$$\mathbb{P}_\xi\left[\text{sign}\left[\sum_{i=1}^{M}\text{sign}(f(x^k + \gamma_k\mathbf{e}^k, \xi_{i,+}^k) - f(x^k - \gamma_k\mathbf{e}^k, \xi_{i,-}^k))\right] \neq \text{sign}(\langle\nabla f(x^k), \mathbf{e}^k\rangle)\right].$$

Each comparison $\text{sign}(f(x^k + \gamma_k\mathbf{e}^k, \xi_{i,+}^k) - f(x^k - \gamma_k\mathbf{e}^k, \xi_{i,-}^k)) \neq \text{sign}(\langle\nabla f(x^k), \mathbf{e}^k\rangle)$ is a Bernoulli trial with failure probability (48):

$$\mathbb{P}_\xi\left[\text{sign}(f(x^k + \gamma_k\mathbf{e}^k, \xi_{i,+}^k) - f(x^k - \gamma_k\mathbf{e}^k, \xi_{i,-}^k)) \neq \text{sign}(\langle\nabla f(x^k), \mathbf{e}^k\rangle)\right] \leq \mathbb{P}_\xi\left[|\theta_{i,+}^k - \theta_{i,-}^k| \geq \gamma_k\cdot|\langle\nabla f(x^k), \mathbf{e}^k\rangle|\right].$$

The right probability can be estimated using Gauss inequality (see Lemma 6 in proof C.5) for unimodal symmetric noise $\theta_{i,+}^k - \theta_{i,-}^k$ by (35) with $S = \frac{|\langle \nabla f(x^k), \mathbf{e}^k \rangle|}{2\sigma}$:

$$\mathbb{P}_\xi \left[ \text{sign}(f(x^k + \gamma_k \mathbf{e}^k, \xi_{i,+}^k) - f(x^k - \gamma_k \mathbf{e}^k, \xi_{i,-}^k)) \neq \text{sign}(\langle \nabla f(x^k), \mathbf{e}^k \rangle) \right] \leq \begin{cases} \frac{1}{2} \left( \frac{\kappa}{\kappa+1} \right)^\kappa \frac{1}{S^\kappa}, & S^\kappa \geq \frac{\kappa^\kappa}{(\kappa+1)^{\kappa-1}}, \\ \frac{1}{2} - \frac{1}{2} \frac{S}{(\kappa+1)^{\frac{1}{\kappa}}}, & S^\kappa \leq \frac{\kappa^\kappa}{(\kappa+1)^{\kappa-1}}. \end{cases}$$

For the probabilities with upper bounds like this, the resulting probability after $M$ Bernoulli trials can be bounded by (See proof C.5 from (35) until (41)):

$$\mathbb{P}_\xi \left[ \text{sign} \left[ \sum_{i=1}^M \text{sign}(f(x^k + \gamma_k \mathbf{e}^k, \xi_{i,+}^k) - f(x^k - \gamma_k \mathbf{e}^k, \xi_{i,-}^k)) \right] \neq \text{sign}(\langle \nabla f(x^k), \mathbf{e}^k \rangle) \right] \leq \left( \frac{\kappa+1}{\kappa-1} \right)^{\frac{1}{\kappa}} \cdot \frac{1}{\sqrt{M}} \cdot \frac{1}{S_j}$$

$$= \left( \frac{\kappa+1}{\kappa-1} \right)^{\frac{1}{\kappa}} \cdot \frac{1}{\sqrt{M}} \frac{2\sigma}{|\langle \nabla f(x^k), \mathbf{e}^k \rangle|},$$

where we denote $a_\kappa := \left( \frac{\kappa+1}{\kappa-1} \right)^{\frac{1}{\kappa}}$. The rest of the proof copies the proof of CompSGD Convergence Lemma C.6 with substitution $\sigma \to \frac{a_\kappa \sigma}{\sqrt{M}}$, and we obtain the bound:

$$\frac{1}{T} \sum_{k=1}^T \|\nabla f(x^k)\|_p \leq \frac{2\Delta_1}{T\alpha_p \gamma} + 24 d^{\frac{1}{p} - \frac{1}{2}} \frac{L\gamma}{\alpha_p} \log(1/\delta) + \frac{16 a_\kappa \sigma}{\sqrt{M} \alpha_p \gamma} + \frac{12 d^{\frac{1}{p} - \frac{1}{2}} \|\nabla f(x^1)\|_2}{T\alpha_p} \log(1/\delta).$$

The last term converges much faster than other, hence, we neglect it. We also use notation $L_{\delta,p} = d^{\frac{1}{p} - \frac{1}{2}} L \log(1/\delta)$.

**1) For arbitrary tuning**, we use parameters $T, \gamma_k = \frac{\gamma_0}{\sqrt{T}}, M_k = \max\{1, M_0 T^2\}$ to get:

$$\frac{1}{T} \sum_{k=1}^T \|\nabla f(x^k)\|_p \leq \frac{2\Delta_1}{\sqrt{T}\alpha_p \gamma_0} + 24 d^{\frac{1}{p} - \frac{1}{2}} \frac{L\gamma_0}{\sqrt{T}\alpha_p} \log(1/\delta) + \frac{16 a_\kappa \sigma}{\sqrt{M_0}\alpha_p \gamma_0 \sqrt{T}} + \frac{12 d^{\frac{1}{p} - \frac{1}{2}} \|\nabla f(x^1)\|_2}{T\alpha_p} \log(1/\delta).$$

Setting such $T$ that the first three terms become less than $\varepsilon$, we obtain the final complexity $N = T \cdot M_0 T^2$.

**2) For optimal tuning**, we first choose large enough $M$ to bound the term $\frac{16 a_\kappa \sigma}{\sqrt{M_k} \alpha_p \gamma} \leq \varepsilon/2 \Rightarrow M_k \equiv \max \left\{ 1, \left( \frac{32 a_\kappa \sigma}{\alpha_p \varepsilon \gamma} \right)^2 \right\}$.

Then we choose optimal $\gamma = \sqrt{\frac{\Delta_1}{12 L_{\delta,p} T}}$ minimizing $\min_\gamma \frac{1}{\alpha_p} \left\{ \frac{2\Delta_1}{T\gamma} + 24 L_{\delta,p} \gamma \right\} = \frac{1}{\alpha_p} \sqrt{\frac{48 \Delta_1 L_{\delta,p}}{T}}$. Finally, $T$ is set to bound $\frac{1}{\alpha_p} \sqrt{\frac{48 \Delta_1 L_{\delta,p}}{T}} \leq \varepsilon/2 \Rightarrow T = O \left( \frac{\Delta_1 L_{\delta,p}}{\alpha_p^2 \varepsilon^2} \right)$. $\qquad\square$

## D. Experimental details

### D.1. LLaMA 130M pre-training on C4

We adopt a LLaMA-based architecture (Touvron et al., 2023) with RMSNorm and SwiGLU (Shazeer, 2020) activations on the C4 dataset (Raffel et al., 2020). Following Lialin et al. (2023), we trained for 100k steps using a batch size of 512 sequences, sequence length of 256. We used T5 tokenizer, since it also was trained on C4 with dictionary size equal to 32k.

For all experiments, while the main model parameters use the respective optimization method, the LM head layer is optimized with AdamW. This follows prior work Zhao et al. (2024) which demonstrated that the LM head layer requires more nuanced effective learning rate adaptation across different tokens for optimal performance. We used Nesterov acceleration scheme with momentum value of 0.9 for all methods except AdamW. For AdamW we used standard hyperparameters: $\beta_1 = 0.9, \beta_2 = 0.999, \varepsilon = 1e - 8$.

We selected the learning rate through a grid search with multiplicative step of $10^{\frac{1}{4}}$. We used a cosine learning rate schedule with a warmup of 10% of the total number of steps and decay of the final learning rate down to 10% of the peak learning rate. For all methods except M-NSGD we used gradient clipping with threshold of 1.0 In addition, we selected best weight decay value between [0, 0.01, 0.1].

## D.2. RoBERTa large fine-tuning

For these experiments, we follow Gao et al. (2020b) for the prompt-based fine-tuning paradigm for masked language models and reuse training hyperparameters from Malladi et al. (2023a). Please refer to the original papers for more details. We compare methods in few-shot scenario with $k = 16$ examples.

For CompSGD Algorithm 5, we sampled $\mathbf{e}^k$ from scaled Euclidian sphere, i.e. $\alpha \cdot S_2^d = \{\mathbf{e} | \|\mathbf{e}\|_2 = \alpha\}$. We set $\alpha$ equal to 17 for all datasets and selected the learning rate in [0.3, 1.0, 3.0] based on validation score.

