# OpenReview forum: "Sign Operator for Coping with Heavy-Tailed Noise: High Probability Convergence Bounds with Extensions to Distributed Optimization and Comparison Oracle"
_ICML.cc/2025/Conference — Submitted to ICML 2025_

### Official Review · Reviewer_5Sbn · 2025-02-16

**Overall Recommendation:** 4

**Summary:**

### **Motivation**
Heavy-tailed noise is a common issue in large-scale machine learning optimization, particularly for LLMs. Approaches like gradient clipping and normalization are common ways to mitigate this issue but require careful tuning or are simply not suitable for distributed settings. This paper explores the use of sign-based optimization as a simple and effective alternative.

### **Key Contributions**
1. Sign-Based Methods for HT Noise
   - The authors demonstrate that SignSGD achieves optimal convergence under HT noise without additional hyperparameters.
   - They provide the first high-probability convergence bounds for SignSGD for non-convex losses.

2. High-Probability Convergence Guarantees
   - SignSGD with minibatching achieves an optimal complexity for HT noise with a bounded $\kappa$-th moment $\kappa \in (1,2]$.
   - SignSGD with Majority Voting further improves convergence under symmetric noise.

3. Extension to Zeroth-Order Optimization
   - The paper introduces MajorityVote-CompSGD, an algorithm for optimization with comparison-based oracles.
   - It provides the first HP bound for zeroth-order optimization under HT noise.

### Highlights
- Sign-based optimization is highly effective in handling heavy-tailed noise.
- Majority Voting significantly reduces noise.
- Sign-based methods outperform traditional approaches, making them particularly useful for LLM training and stochastic optimization.

# Update After Rebuttal

Dear Authors,

You have shown great effort, and I am satisfied with your reply.

I am sure you will:

1. Include your newly found results and compare them with the relevant literature ([1,2,3,4]).
2. Include your new experiments to validate your bounds. These can still be improved, e.g., larger and different settings, but they are a step in the correct direction.
3. Rectify your claims about your methods being somewhat uniformly better than AdamW or so, and rather being more precise about this type of claim.

I appreciate your reply: I will follow the discussion with the other Reviewers. For the moment, I decided to raise my score to Accept (**4**), but **not** to Strong Accept (**5**).

**Claims And Evidence:**

While the theory looks sound, not all the theoretical insights are necessarily validated appropriately in meaningful setups. More details later.

**Essential References Not Discussed:**

See above.

**Experimental Designs Or Analyses:**

As I discussed above, I believe the experimental setup is too limited to draw any meaningful conclusion. In LLMs, scale is THE challenge and different (and larger) scales bring different challenges. Not testing the behavior at different increasing scales of a model is not enough to claim that "Our theoretical findings are supported by the superior performance of sign-based methods in training Large Language Models".

**Methods And Evaluation Criteria:**

I honestly do not think the evaluation criteria are sufficient: Claiming that an optimizer is **better than AdamW** for LLMs is among the strongest claims one can make in this field. I believe training on such a small LLM is by no means representative, but certainly encouraging.

Claiming to be better than AdamW requires extensive evaluations across multiple architectures and datasets. A good start could be validating this claim on "Pythia: A Suite for Analyzing Large Language Models Across Training and Scaling". And still, this would not be enough to say your algo is better than AdamW, but it would be a much more solid basis to say it is promising.

**Other Comments Or Suggestions:**

# Conclusion

From a theoretical standpoint, the paper seems solid and simply needs a bit of contextualization w.r.t. recent literature. The results are novel and the new optimizers show promising results from an experimental point of view.

**However**, the experimental evaluation is too restrictive to draw any meaningful conclusion w.r.t. whether or not these new optimizers truly outperform AdamW. As this claim would have major repercussions for the DL community, it has to be thoroughly verified.

**IF** you can show that your optimizers outperform AdamW (ofc AdamW would need first to be tuned to find **optimal parameters**!) on a variety of architectures and datasets (see at least my suggestions above regarding Pythia), this paper deserves a **Strong Accept**.

For the time being, I think it is fair to give **Weak Accept** and I will follow the discussion during the rebuttal time: I see potential to give **Accept**.

**Other Strengths And Weaknesses:**

The main issue is truly the experimental side, which I discussed above. To build on that, I believe that it could be very educational to verify the bounds empirically on some simple settings: A straightforward approach to address this would be to train a small model—such as a shallow MLP—while injecting noise into the full gradient. This setup does not require a large-scale LLM while it allows for systematic testing of the theoretical predictions.

By varying each parameter appearing in the derived bounds, one could assess whether the theoretical results align with empirical observations.

For instance, training a small MLP while injecting noise of magnitude $\sigma$ could verify whether the L1 norm of the gradient truly scales linearly as predicted by Lemma 1. Similarly, other key hyperparameters could be ablated.

Such an evaluation would give a bit of intuition on how tight the analysis is and how loose the bounds are.

**Questions For Authors:**

Can you somehow show linear speed-up in terms of the number of clients for your distributed optimizer?

**Relation To Broader Scientific Literature:**

While browsing for recent literature, I came across two concurrent works [1,2] that were recently published --- They contain some interesting results that deserve some comments as they are very much related to those derived in this paper:

[1] Derives a convergence bound for SignSGD under Gaussian and Student's t gradient noise when the variance of the noise is unbounded;

[2] Generalizes [1] to the distributed setting and studies also the case where the *expected value* of the noise is unbounded: Distributed SignSGD seems to converge even in this case and it is empirically verified across different architectures and datasets.

I suggest:

1. Discussing results like Lemma 3.5 in [1] or Theorem 3.12 and Theorem 3.13 in [2] regarding the role of $\sigma$ and comparing these with the bounds derived in this paper.
2. Commenting on why the analysis in this paper requires the expected value of the noise to be bounded, while this is not necessarily necessary in Theorem 3.12 and Theorem 3.13 of [2] with $\nu=1$.


[1] "Adaptive Methods through the Lens of SDEs: Theoretical Insights on the Role of Noise", Compagnoni et al. ICLR 2025.

[2] "Unbiased and Sign Compression in Distributed Learning: Comparing Noise Resilience via SDEs", Compagnoni et al. AISTATS 2025.

**Theoretical Claims:**

No, I did not check the proofs in detail: I skimmed over the main results and they seem fine. The results themselves make sense in light of the existing literature.

---

> ### Author Rebuttal · Authors · 2025-04-01
>
> Dear Reviewer 5Sbn, thank you for your questions and positive evaluation of our paper.
>
> **Claiming that an optimizer is better than AdamW.**
>
> By saying this in abstract, we did not mean to suggest that the sign-based method is the best possible method for language model pretraining, but rather that **it can compete with other methods commonly considered in theory for dealing with heavy tails** -- specifically, clipping and normalization. We sincerely apologize if our wording misled the Reviewer.
>
> We also do not claim that M-SignSGD is a new SOTA uniformly surpassing AdamW across a wide range of practically important tasks. However, our experiments, along with the results from [Zhao 2024], demonstrate that it can at least provide serious competition to well-tuned AdamW at small scales (up to 1B) of dense language model pretraining.
>
> To further explore the potential applicability of M-SignSGD in practically important tasks, **we conducted additional experiments** with the pretraining of a Mixture-of-Experts language model of larger size. The results can be found in **reply 1 for Reviewer AScN.**
>
> [Zhao 2024] Zhao, R. et al. Deconstructing what makes a good optimizer
> for language models, 2024.
>
>
> **Related works.** Indeed, in works [1] and [2], the authors work with Student's distributions which can have unbounded math expectation. Symmetric noises can break traditional lower bounds and guarantee convergence when noises have bounded $\kappa$-th moment, $\kappa > 0$, see [3, 4]. Therefore, for a fair comparison, we provide our rates for MajorityVote-SignSGD $\kappa \in (1,2]$ (Theorem 3):
>
> $$N = O\left(\frac{ \Delta_1 L_\delta d}{\varepsilon^2} +   \frac{ \Delta_1L_\delta d} { \varepsilon^2}
>  \left(\frac{a_\kappa||\sigma||_1}{ \varepsilon}\right)^2\right),$$
>
> where $(a_\kappa)^\kappa := \frac{\kappa+1}{\kappa - 1}$.  The dependence on $\kappa$ is only in a deteriating factor $a_\kappa$. We did not consider the case $0 < \kappa \leq 1$, however, after careful check, we proved that MajorityVote-SignSGD can work with any $\kappa >0$.  We managed to eliminate the factor $a_\kappa$ and obtain high probability rates with $M = \max\left(\frac{(\kappa + 1)^{2\kappa - 1}}{\kappa^{\kappa + 2}}, (\frac{||\sigma||_1}{\varepsilon})^2 \right)$:
>
> $$N = O\left(\frac{ \Delta_1 L_\delta d}{\varepsilon^2\kappa^4} +   \frac{ \Delta_1L_\delta d} { \varepsilon^2}
>  \left(\frac{\|\sigma\|_1}{ \varepsilon}\right)^2\right).$$
>
> Hence, **MajorityVote-SignSGD  can also operate under $1 \geq \kappa > 0$ and symmetric HT noise** similar to the DSignSGD from [2]. We will add case $\kappa > 0$ in a revision.
>
> In our bounds, the dependence on $\sigma$ is quadratic, and, in [2], the dependence is linear. We are not aware of any lower bounds for **symmetric** HT noises. Thus, we cannot say whether the linear dependence on $\sigma$ is optimal or not. The linear dependency in [2] could have emerged since the authors consider only Student's distributions.
>
>
> [1] "Adaptive Methods through the Lens of SDEs: Theoretical Insights on the Role of Noise", Compagnoni et al.
>
> [2] "Unbiased and Sign Compression in Distributed Learning: Comparing Noise Resilience via SDEs", Compagnoni et al.
>
> [3] Puchkin, N.et al. Breaking the heavy-tailed noise barrier in stochastic
> optimization problems.
>
> [4] Armacki, A.et al. Large deviations and improved mean-squared error
> rates of nonlinear sgd: Heavy-tailed noise and power of
> symmetry.
>
> **Validate theory in practice.**
> Following your advice, we trained MLP on 'mushrooms' dataset to practically validate the dependence on noise.  Lemma 1 states that SignSGD without batching or voting converges with rate
>
> $$\frac1T \sum\limits_{k=1}^{T} \|\nabla f (x^k)\|_1 \leq \frac{2\Delta_1}{T\gamma} + 16 Ld\gamma \log(\frac{1}{\delta})  + 4 \sigma \notag +  12\frac{d\|\nabla f (x^1)\|_1}{T}  \log(\frac{1}{\delta}).$$
>
> Figure 1: https://limewire.com/d/llU7f#n4qOoELVLI
> Figure 2: https://limewire.com/d/xf3xK#0fFssBjtev
> Figure 3: https://limewire.com/d/lm6Z3#CfR5OGQit4
>
> If one takes a small batchsize $\gamma$ and a large number of iterations $T$, then method reaches $\sigma$ plateau despite $\kappa$. In Figure 1, we did this experiment, and one can see that plateau level in practice linearly depends on $\sigma$ and does not depend on $\kappa$.
>
> Theorem 3 and the updated rates from the above question state that batchsize $M$ in MajorityVote-SignSGD reduces achieved plateau level as $\frac{\sigma}{\sqrt{M}}$ for any $\kappa \in [1,2].$ In Figure 2, we took quadratically growing batchsizes $[1, 5, 17]$ and fixed $\sigma$, and plateau level actually drops linearly despite $\kappa$ (even $\kappa = 1$).
>
> Similarly, according to Theorem 2, simple batching theoretically drops plateau level as $\sigma/B^{\frac{\kappa - 1}{\kappa}}$. In Figure 3, we verified it for $\kappa = 2$ with quadratic batchsizes and for $\kappa = 1.5$ with cubic batchsizes.
>
> **Therefore, we demonstrate that our theoretical bounds seem to be tight in practice.**

---

> > ### Comment · Reviewer_5Sbn · 2025-04-01
> >
> > Dear Authors,
> >
> > you have shown great effort, and I am satisfied with your reply.
> >
> > I am sure you will:
> >
> > 1. Include your newly found results and compare them with the relevant literature ([1,2,3,4]).
> > 2. Include your new experiments to validate your bounds. These can still be improved, e.g., larger $T$, and different settings, but they are a step in the correct direction.
> > 3. Rectify your claims about your methods being somewhat uniformly better than AdamW or so, and rather being more precise about this type of claim.
> >
> > I appreciate your reply: I will follow the discussion with the other Reviewers. For the moment, I decided to raise my score to Accept (**4**), but **not** to Strong Accept (**5**).

---

### Official Review · Reviewer_1isr · 2025-03-10

**Overall Recommendation:** 1

**Summary:**

In this paper, the authors consider the problem of minimizing a non convex function under two different oracle models. In the first case, the algorithm access to a noisy first order oracle obeying to an heavy-tailed distribution. In this model, authors analyze a number of version of the sign SGD method, and prove several sample complexity bounds. In the second model, the oracle returns zeroth order information corrupted by heavy tailed noise. Even in this case, authors consider a number of variants of sign SGD-like methods for which they prove sample complexity bounds. Preliminary experiments are provided.

## Update after rebuttal ##
I thank the authors for the rebuttal, however I'll keep my score due to the following reasons:

The authors have overlooked an important related work in their discussion and appear (perhaps unintentionally) biased toward earlier contributions in this area. The clarifications provided in the rebuttal should have been included in the original submission.

Some contributions seem overstated. For instance, the claim that the numerical constants for sign SGD are smaller than those of alternative methods is not particularly meaningful. Since prior works did not focus on optimizing constants, and the minimax values are unknown, comparing constants in upper bounds across different methods lacks justification. Such a comparison would only be appropriate if the authors had improved the upper bound of an existing method — which is not the case here.

Overall, the paper feels somewhat rushed in its current form. Authors use a somehow sloppy jargon in several point of the manuscript, as I highlighted in my review.

Technical novelty is limited. This is not a major problem, as long as the results are interesting, but combined with a poor presentation and discussion of the related work, contribute to my low score.

Sign SGD is an interesting and robust alternative to standard SGD. In that respect, this paper addresses a relevant problem. However, I believe that the present submission requires a major revision — not just a camera-ready version — incorporating the points raised in the rebuttal and presenting the material in a more polished and thorough manner, before it is ready for publication.

**Claims And Evidence:**

See discussion below.

**Essential References Not Discussed:**

See discussion below.

**Experimental Designs Or Analyses:**

See discussion below.

**Methods And Evaluation Criteria:**

See discussion below.

**Other Comments Or Suggestions:**

In section 2.1. authors may add a comment on how their component wise assumption on the noise relate to the canonical norm assumption on the oracle.

In section 2.4, paragraph Majority Voting, the $g_i$ are used without begin introduced.

Please rephrase this sentence: *Choosing the most frequent value from the sign...M Bernoulli trials.*

There is a typo in line 2 of Algorithm 3, $B_k$ should be replaced by $M_k$.

In section 3.1 it should be $\mathbb{R}^d$ not $\mathbb{R}^D$.

**Other Strengths And Weaknesses:**

The paper consider a problem which is relevant for the ICML community.

**Writing:**
The overall writing could be improved as currently it appear to be a bit rushed. In several point, authors use a rather informal prose, and some effort is required to put claims in context. As an example, I found Remark 2 quite confusing: what the authors mean by "the batched function values"? More examples of inaccurate claims or phrases will follow.

**References:** The related work section for the first order method is lacking some important references and is inaccurate.

For clipping on non-convex objective, the authors do not cite

[Nguyen2023] = *Improved Convergence in High Probability of Clipped Gradient Methods with Heavy Tailed Noise*. Nguyen et al. NeurIPS 2023.

In that paper authors prove rates of convergence (for the average gradient norm), in both the finite horizon (known number of iteration) and the any-time setting (unknown number of iteration), that are comparable, or better, to those prove in this paper.

In the finite horizon setting, the authors show a rate of (the order of) $T^{(2\kappa-2)/(3\kappa-2)}$ with unit batch-size. This leads to a sample complexity of order

$\bigg(\frac{1}{\varepsilon}\bigg)^{\frac{2\kappa-2}{3\kappa-2}}$

Notice that this is better of the results in (3) and (4) of the present paper . This rate is also better then that in (7) and (8) for every $\kappa < 6/5$.

In addition, authors of it is worth noting that authors of [Nguyen2023] also consider the any-time setting, where they show similar results up to the introduction of additional logarithmic terms in $1/\varepsilon$.

Finally, in the same work, Nguyen et al. also prove results for the convex setting, which the author of this works seems to not consider.

**Inaccurate claims/Clarifications:** In section 1.1., clipping paragraph, authors mention a lower bound by Zhang et al. it would be useful to state the lower bound or include a table with the most important sample complexity bounds.

In the same paragraph, and in other places in the paper, authors say that symmetric noise is a relaxation of the Heavy-Tailed noise. This is not true, not all heavy-tailed distribution are symmetric (e..g, the Pareto). Similarly there are symmetric distribution that do not have finite expectation. Thus the two classes are not in an inclusion relation. Authors should clarify what they mean by relaxation.

Still in section 1.1. authors say that Liu et al. 2023, and Cutkosky & Metha 2021 obtains high probability bounds under heavy tailed noise  for normalized SGD. How those bounds compare with that obtained in this paper?

Authors define arbitrary the first type of tuning in all their results, but this tuning leave only a constant parameter free to the user. A real free tuning would have left the choices of the step-size and the batch-size completely free.

Below Theorem 3, authors claim that:*In comparison with (4) for minibatch-SignSGD, in expectation....but they are still close due to the norm relation (10).* I'm not sure to understand that, there may still be a factor $\sqrt{d}$ between these quantities, and that can be large in high dimension.

The discussion in section 2.6. is inaccurate. Comparing with clipping authors ignore the work of [Nguyen2023], and only refer to Sadiev et al. 2023. Even limiting to this latter work, one should notice that the authors do not make effort in optimizing the constants, and therefore it cannot be concluded that clipping suffers from worse constants...

The parameter setting proposed by the authors for their variants of sign SGD also on the number of iterations, and then it is unclear what do they mean by *The sign-based methods work well with constant, arbitrary parameters.*

**Conclusion:** The problem faced in this work is interesting, but I think that a major rework is needed to include a more comprehensive discussion of the related works (including the related paper [Nguyen2023]), a more extensive comparison with the existing results, and a more careful writing and presentation of the results.

**Questions For Authors:**

In section 1.2, first bullet point, authors says that they generalize their high probability bound to a bound in expectation. In which sense, their in expectation bound is a generalization?

In eq. (3) where is the dependence on $\delta$?

Below eq. (5) authors claim that it is not possible to obtain decreasing probability for $\sign(\nabla f(x^k))_j \neq \sign(\sum_{i=1}^M \sign(g_i^k)_j$, unless one place additional assumptions. Why is that?

Could you add a reference for this claim? *For example, the condition (6) is satisfied if the noise of the gradient estimate for each coordinate is unimodal and symmetric about its mean.*

In the experimental section, it is unclear to me how many repetitions have been performed in each experiment and what statistics authors reported. Could you clarify it?

**Relation To Broader Scientific Literature:**

See discussion below.

**Theoretical Claims:**

See discussion below.

---

> ### Author Rebuttal · Authors · 2025-04-01
>
> Dear Reviewer 1isr, thank you for your valuable feedback.
>
> We sincerely apologize for overlooking certain important works. We will address these gaps in a revision.  Nevertheless, we would like to emphasize that our main results remain competitive with or outperform mentioned methods. In this case, suggested presentation, typos and unclear moments can be quickly fixed.
>
> **[Nguyen2023]** The rates from [Nguyen2023] are given for the average SQUARED $l_2$-norm, i.e., to achieve $\| \nabla f(x)\|^2_2 \leq \varepsilon^2$ one needs
> $$\max \left(\left(\frac{8 ||\sigma||_2^\kappa \log \frac{1}{\delta}}{\sqrt{\Delta_1 L}} \right)^\frac{3\kappa -2}{\kappa - 1} \left(\frac{720 \sqrt{\Delta_1 L}\log \frac{1}{\delta} }{\varepsilon^2}\right)^\frac{3\kappa - 2}{2\kappa - 2}, \left(\frac{720 \cdot 32^\frac1\kappa ||\sigma||_2\log \frac{1}{\delta} }{\varepsilon^2}\right)^\frac{3\kappa - 2}{2\kappa - 2}, \left(\frac{1440\sqrt{90} \cdot L\Delta_1 \log \frac{1}{\delta} }{\varepsilon^2}\right)^\frac{3\kappa - 2}{2\kappa - 1}\right).$$
> For the minibatch-SignSGD, the bound (4) is given in terms $\| \cdot \|_1 \leq \varepsilon$ (WITHOUT SQUARE):
> $$ \max \left(\frac{512 \Delta_1 L \log \frac{1}{\delta} d }{\varepsilon^2},   \frac{512 (16 ||\sigma||_1)^\frac{\kappa}{\kappa -1} \Delta_1 L \log \frac{1}{\delta} d }{\varepsilon^\frac{3\kappa -2}{\kappa - 1}} \right)   \quad \quad (*).$$
> Our bound has the same dependency on $\varepsilon$ for $\forall \varepsilon, \sigma > 0$ and even better one if $\sigma = 0$. The dependencies on $L,\sigma$ are much closer to the lower bound $\Omega \left( \frac{\Delta_1 L}{\varepsilon^2} + \frac{\Delta_1 L||\sigma||_2^\frac{\kappa}{\kappa - 1} }{\varepsilon^\frac{3\kappa -2}{\kappa - 1}} \right)$ from [Zhang2020]. Finally, the numerical constants with $\log \frac1\delta$ factors are milder in our bounds.
>
> **Additional $\sqrt{d}$ factors** are inherent to sign methods, since they work with larger $l_1$ norm instead of $l_2$. We can derive bound  $||\cdot||_2 \leq \varepsilon$ with the following density function $\phi(u) := \frac{||u||_1}{\sqrt{d}||u||_2}$ ($= 1$ for dense vectors and $=\frac{1}{\sqrt{d}}$ for sparse):
>
> $$O\left( \frac{\Delta_1 L_\delta } {\varepsilon^2\phi^2(g)}
>  \left( 1 + \frac{||\sigma||_2 \phi(\sigma)}{\varepsilon \phi(g)}\right)^\frac{\kappa}{\kappa-1}\right),$$
> where $\phi(g)$ is the average density of gradients. In practice, this density remains close to $1$ (see [1] (Section 3)), therefore, high dimensionality does not immediately imply slower convergence.
>
> **Convex case.** We left the convex case out of the scope of our work, since its primal goal is to effectively and robustly solve non-convex ML and DL tasks under HT noise. The sign-based methods can work with  convexity (see, [2]).
>
> **Infinite horizon.** Sign methods can work with infinite horizon ([3] Theorem 1). For minibatch-SignSGD, we can provably take stepsizes $\gamma_k \sim 1/\sqrt{k}$ and  $B_k \sim  k^\frac{\kappa}{2\kappa-2}$  to obtain rates with extra $\log T$ factors and the same sample complexity to get $\min_k \|\nabla f(x^k) \|_1  \leq \varepsilon$.
>
> **Clip Normalized SGD.** In [Cutkosky Metha], the rates are obtained for the bounded gradient $\kappa$-th norm, and [Liu] generalizes it for the bounded  $\kappa$-th moment:
> $$O\left(\frac{\sqrt{L\Delta_1}}{T^\frac{\kappa - 1}{3\kappa - 2}} + \frac{\sigma\log T/\delta}{T^\frac{\kappa - 1} {3\kappa - 2}}\right).$$
> Compared to ours, these rates have extra $\log T$ factors and are not optimal for $\sigma = 0.$
>
> **Complexity table.** For the sake of readability, we put lower bound in Section 2.6 line 272, where we discuss explicit formulas.
>
> **Clipping Constants.** We do not optimize constants as much as possible. However, constants in bound (*) are clearly better than the one from [Nguyen2023]. Our constants follows from proof lines 810-850.
>
> **Parameter free.** We did not try to build parameter-free methods. We only mean that even without knowing anything about the problem, one can expect and evaluate convergence of sign-based methods using the proposed problem-agnostic parameters.
>
> **Word "generalization".** The word "generalization" is incorrect for in expectation bounds and symmetric HT noises. We will replace it.
>
> **Eq3.** There must be $L_\delta$ instead of $L$.
>
> **Below eq. (5)** There is a $1$-d counterexample: $\nabla f = 0.1$ and noise $\xi = -1$ with $99\%$ and $\xi=1$ with $1\%$. The probability of getting the right sign with 1 trial is $0.01$, and with $3$ trials $\leq 2 * 0.01 * 0.01$.
>
> **Reference for this claim?** It was proved for BV noise in [1](Theorem 2). For HT noise, we prove it in the beginning of Section C.5.
>
>
> **Experimental section**
>
> We ran all experiments with 3 different random seeds. For pretraining, we report validation perplexity, and for fine-tuning, we report target task test accuracy.
>
> **References:**
>
> [1] https://arxiv.org/pdf/1802.04434
>
> [2] https://arxiv.org/abs/1901.09847
>
> [3] https://arxiv.org/abs/1905.12938

---

### Official Review · Reviewer_AScN · 2025-03-18

**Overall Recommendation:** 3

**Summary:**

This paper investigates the robustness of a series of sign-based stochastic optimization methods for handling heavy-tailed (HT) noise in smooth non-convex functions. The authors argue that leveraging the sign of gradient estimates, without introducing additional hyperparameters, effectively addresses HT noise. The key contributions include:

1. A family of methods including SignSGD, SignSGD with Majority Voting, and a novel zeroth-order method, MajorityVote-CompsSGD;

2.  High-probability (HP) sample complexity bounds for these methods under HT noise, such as $\tilde{O}\left(\varepsilon^{-\frac{3\kappa - 2}{\kappa - 1}}\right)$ for SignSGD to achieve gradient norm accuracy $\varepsilon$;  $\tilde{O}(\varepsilon^{-4})$ for SignSGD with Majority Voting under symmetric noise and $\tilde{O}(\varepsilon^{-6})$ for MajorityVote-CompsSGD in terms of comparison

3. Experimental validation on Transformer models (LLaMA pre-training and RoBERTa fine-tuning). The paper claims these methods collectively outperform clipping-based approaches by avoiding complex tuning.

**Claims And Evidence:**

The claims are generally supported by theoretical proofs and experimental results. The HP bounds for SignSGD (Theorem 1) and its variants (Theorems 2-4) are derived under clear assumptions (smoothness, lower boundedness, and HT noise with bounded κ-th moment), with proofs deferred to Appendix C. Experimental validation on Transformer models provides evidence of practical effectiveness.

**Essential References Not Discussed:**

N/A

**Experimental Designs Or Analyses:**

The experiments involve pre-training LLaMA 130M on the C4 dataset and fine-tuning RoBERTa on NLP tasks.

The design is sound, targeting real-world scenarios.

Metrics (e.g., gradient norm reduction, task performance) are not fully specified, which reduces transparency.

**Methods And Evaluation Criteria:**

The proposed methods—SignSGD, SignSGD with Majority Voting, and MajorityVote-CompsSGD are well expressed.

The evaluation criteria. The use of Transformer models (LLaMA pre-training and RoBERTa fine-tuning) as benchmarks is appropriate given their relevance to modern machine learning tasks. Expanding the evaluation with additional benchmarks could further enrich the findings and enhance their generalizability.

**Other Comments Or Suggestions:**

Can you provide a direct experimental comparison with ClipSGD under HT noise to substantiate the superiority claim?
This could strengthen my confidence in the practical advantage.

**Other Strengths And Weaknesses:**

N/A

**Questions For Authors:**

Can you provide a direct experimental comparison with ClipSGD under HT noise to substantiate the superiority claim?
This could strengthen my confidence in the practical advantage.

**Relation To Broader Scientific Literature:**

This paper presents work whose goal is to advance the field of Machine Learning. None of them must be specifically highlighted here.

**Theoretical Claims:**

I reviewed the convergence theorems for the main proposed methods and their assumptions, as outlined in Section 2.

The theorems' high-probability (HP) bounds appear to be correctly derived under the stated assumptions, balancing terms such as the smoothness constant L, noise magnitude σ, and failure probability δ.

While the proofs are not included in the main text, the assumptions (e.g., L-smoothness, bounded κ-th moment noise) are standard and reasonable.

I did not detect any obvious errors in the formulation, though I could not fully verify the proofs without access to Appendix C.

---

> ### Author Rebuttal · Authors · 2025-04-01
>
> Dear Reviewer AScN, thank you for your time and comments and positive evaluation of our paper:
>
> **(1) Expanding the evaluation with additional benchmarks could further enrich the findings and enhance their generalizability.**
>
> Thank you for raising this point.
> We complement our experiments with a new setup -- new architecture and data.
> Previously, we used a dense LLaMA model; now, we have switched to a Mixture of Experts (MoE) architecture based on the same LLaMA model, retaining RoPE and identical activation functions. Our MoE model follows the Switch Transformer [1] MoE variant with classical top k=2 gating and 8 experts, giving us approximately 520M parameters if we have the same configuration as 130M LLaMA. We conduct this experiments on the FineWeb dataset [1] a popular corpus for LLM pre-training.
>
> We run AdamW, M-SignSGD, M-NSGD and M-ClippedSignSGD optimizers following the best practices from our earlier setup on dense models. We train with a batch size of 256 and sequence length 512 for 42k (5.5B tokens)and 336k steps (44B tokens). That is for the second training horizon we go far beyond the Chinchilla optimal tokens-per-parameters ration.
>
> Perplexity of LLaMa-base MoE 520M model pre-trained on FineWeb for 42k steps. Lower is better
>
> AdamW:  ppl 22.85
>
> M-SignSGD: ppl 23.19
>
> M-NSGD: ppl 23.32
>
> M-ClippedSignSGD: ppl 23.30
>
> Perplexity of LLaMa-base MoE 520M model pre-trained on FineWeb for 336k steps. Lower is better
>
> AdamW:  ppl 18.68
>
> M-SignSGD: ppl 18.87
>
> We would like to highlight that M-SignSGD scales remarkably well with increasing model size, outperforming M-NSGD and M-ClippedSignSGD. Additionally, we encountered difficulties running M-ClippedSGD in this setting. Consequently, we decided to include a clipped version of M-SignSGD, which aligns with our approach since we consider only an EMA of momentum in the update.
>
> [1] https://arxiv.org/abs/2101.03961
>
> [2] https://arxiv.org/abs/2406.17557
>
> **(2) Metrics (e.g., gradient norm reduction, task performance) are not fully specified, which reduces transparency.**
>
> We thank the Reviewer for this note. Although we specify in the captions of Tables 1 and 2 the metrics being reported, the main text would benefit from a more detailed description of the target metrics. We will be sure to include this in a reversion.
>
> **(3) Can you provide a direct experimental comparison with ClipSGD under HT noise to substantiate the superiority claim? This could strengthen my confidence in the practical advantage.**
>
> Could you, please, clarify what kind of experiments and methods you would like to see? In the reply **(1)**, we conducted additional experiments with LLMs, comparing M-ClipSGD and M-SignSGD methods.

---

> > ### Comment · Reviewer_AScN · 2025-04-04
> >
> > Thank you for your rebuttal; it cleared up my doubts.

---

### Official Review · Reviewer_nio3 · 2025-03-22

**Overall Recommendation:** 3

**Summary:**

This paper studies optimization under heavy-tailed noise. The authors assume that the noise affecting gradient estimates and function evaluations has bounded $\kappa$-th moments for (1,2]. This assumption is common in many large-scale deep learning scenarios. Under this assumption, they develop sign-based methods for both first-order and zeroth-order optimization. In the first-order setting, they introduce variants of SignSGD that incorporate momentum and majority voting. They provide high-probability convergence guarantees for these methods under standard assumptions such as L-smoothness and bounded-below objectives. In the zeroth-order setting, where only comparison feedback is available, they propose CompSGD and an improved version called Majority Vote-CompSGD. Under similar noise and random direction assumptions, they derive high-probability convergence results for these methods. The paper also includes experiments on large-scale language model pretraining and NLP fine-tuning tasks, as well as experiments on synthetic benchmarks, to demonstrate the practical behavior of the proposed algorithms.

**Claims And Evidence:**

The submission presents several claims about convergence under heavy-tailed noise, including high-probability convergence for sign-based first-order methods and comparison-based zeroth-order methods. The authors offer experiments on large-scale language model pretraining, NLP fine-tuning tasks, and synthetic benchmarks that appear to match the theoretical outcomes.

**Essential References Not Discussed:**

NA

**Experimental Designs Or Analyses:**

The experimental designs and analyses, including those for the large-scale language model pretraining, NLP fine-tuning tasks, and synthetic benchmarks for the zeroth-order methods, appear standard for the problem at hand.

**Methods And Evaluation Criteria:**

The proposed methods and evaluation criteria are appropriate for the problem at hand. The authors develop sign-based optimization techniques designed for heavy-tailed noise scenarios and support their analysis with both theoretical derivations and practical experiments. For example, they test their algorithms on large-scale language model pretraining and NLP fine-tuning tasks, which are contexts where heavy-tailed noise is a recognized challenge. In addition, synthetic benchmarks are used to validate the theoretical claims under controlled conditions. Overall, the methods and evaluation criteria are reasonable for addressing robust optimization in heavy-tailed noise settings.

**Other Comments Or Suggestions:**

NA

**Other Strengths And Weaknesses:**

The biggest weakness of the paper is that it mainly integrates existing techniques and extends them under already established assumptions. The work does not introduce fundamentally new ideas, and its contribution in terms of originality is somewhat incremental.

**Questions For Authors:**

In Table 1, M-SignSGD is performing better than AdamW. Do you think this is a hyperparameter tuning issue or M-SignSGD is inherently better for pre-training? Also for paper like Galore[1], the reported perplexity for 130m models are around 25 (see Table 2 in [1]). How come the reported numbers here are so low? Seems that it's due to a larger number of training steps?


References:

[1] Zhao, Jiawei, et al. "Galore: Memory-efficient llm training by gradient low-rank projection." arXiv preprint arXiv:2403.03507 (2024).

**Relation To Broader Scientific Literature:**

The paper draws on multiple threads in the literature related to gradient clipping (Pascanu et al., 2013; Goodfellow et al., 2016), normalization-based SGD (Hazan et al., 2015; Liu et al., 2023; Cutkosky & Mehta, 2021), and sign-based methods (Bernstein et al., 2018a). Prior works often examined these techniques under bounded-variance or symmetric noise assumptions, sometimes offering only expectation-based results. The current submission places these methods in a setting with heavy-tailed noise, which is handled via sign operations, clipping, or normalization. For zeroth-order methods, it references ideas such as the Three-Point method (Bergou et al., 2020; Gorbunov et al., 2022) and noisy comparison oracles (Saha et al., 2021; Lobanov et al., 2024a), extending them to heavy-tailed scenarios with high-probability guarantees. The paper integrates these methods for both first-order and zeroth-order cases, offering a unified approach to robust optimization under heavy-tailed noise.

**Theoretical Claims:**

I did not carefully check the proofs but I believe the proof methodologies are known and should be correct in general.

---

> ### Author Rebuttal · Authors · 2025-04-01
>
> Dear Reviewer nio3, thank you for your feedback and positive evaluation of our paper.
>
> **The biggest weakness of the paper is that it mainly integrates existing techniques and extends them under already established assumptions. The work does not introduce fundamentally new ideas, and its contribution in terms of originality is somewhat incremental.**
>
> Research that provides new insights into already known methods is just as valuable to the community as other types of contributions. For instance, we provide a list of A* conference papers dedicated to the upper bounds for the existing methods:
>
>  [1] Nguyen, T. D., Nguyen, T. H., Ene, A., & Nguyen, H. (2023). Improved convergence in high probability of clipped gradient methods with heavy-tailed noise. Advances in Neural Information Processing Systems, 36, 24191-24222.
>
>  [2] Sun, T., Wang, Q., Li, D., & Wang, B. (2023, July). Momentum ensures convergence of signsgd under weaker assumptions. In International Conference on Machine Learning (pp. 33077-33099). PMLR.
>
>  [3] Sadiev, A., Danilova, M.,  ... & Richtárik, P. (2023, July). High-probability bounds for stochastic optimization and variational inequalities: the case of unbounded variance. In International Conference on Machine Learning (pp. 29563-29648). PMLR.
>
>
> Furthermore, in the context of zeroth-order optimization, we propose novel methods together with their analysis approach via the sign operator. This approach helps uncover and explain the inherent robustness of these methods against noise.
>
>
> **(2) In Table 1, M-SignSGD is performing better than AdamW. Do you think this is a hyperparameter tuning issue or M-SignSGD is inherently better for pre-training?**
>
> We carefully tuned the hyperparameters (see Appendix D.1 for details) and believe that the results for all algorithms in Table 1 are close to optimal. Moreover, similar results were also obtained in the study [4] (see, e.g., Figure 1).
>
> We do not claim that M-SignSGD is inherently better than AdamW for language model pretraining. However, we believe that M-SignSGD can be a serious competitor to AdamW, at least for smaller model sizes.
>
> To further explore the potential applicability of M-SignSGD in practically important tasks, **we conducted additional experiments** with the pretraining of a Mixture-of-Experts language model of larger size. The results can be found in **reply 1 for Reviewer AScN.**
>
> [4] Zhao, R., Morwani, D., Brandfonbrener, D., Vyas, N., and
> Kakade, S. Deconstructing what makes a good optimizer
> for language models, 2024.
>
> **(3)  Also for paper like Galore[1], the reported perplexity for 130m models are around 25 (see Table 2 in [1]). How come the reported numbers here are so low? Seems that it's due to a larger number of training steps?**
>
> As stated in Appendix D.1, we trained the model for 100k steps, unlike the 20k steps in GaLore [5]. Although the number of optimization steps in GaLore aligns with the empirical rule for scaling laws [6], the model remains far from convergence. Since, in practice, models are typically trained on significantly more tokens than suggested by scaling laws, we decided that longer training periods would be more representative.
>
> Moreover, since GaLore explores a memory-efficient setup, the authors trained the model in a pure bfloat16 format (meaning master weights are stored and updated in bfloat16) instead of the standard mixed precision approach for language model pretraining where master weights are kept in float32. This also degrades the model's performance.
>
> [5] Zhao, J., Zhang, Z., Chen, B., Wang, Z., Anandkumar, A., and Tian, Y. Galore: Memory-efficient llm training by gradient low-rank projection, ICML 2024.
>
> [6] Hoffmann, J., Borgeaud, S., Mensch, A., Buchatskaya, E., Cai, T., Rutherford, E., Casas, D. d. L., Hendricks, L. A., Welbl, J., Clark, A., et al. Training compute-optimal large language models, 2022.

---

### Decision · Program_Chairs · 2025-05-01

**Decision:**

Reject

**Comment:**

This paper provides an analysis of optimization with heavy-tailed gradient estimators with symmetric noise. There are several theoretical results suggesting that variants of signSGD work well in this setting, backed up by some basic empirical evidence. The reviews on this paper were rather divergent, but on the whole it seems that there is some concern that the theoretical results are somewhat incremental over prior literature and that the paper would benefit from a more careful presentation.